# A coarse pixel scale ground "truth" dataset based on the global *in situ* site measurements to support validation and bias correction of satellite surface albedo products

Fei Pan[1], Xiaodan Wu[2], Qicheng Zeng[1], Rongqi Tang[1], Jingping Wang[1], Xingwen Lin[3], Dongqin You[4],
Jianguang Wen[4,5], and Qing Xiao[4,5]

[1]College of Earth and Environmental Sciences, Lanzhou University, Lanzhou 730000, China
[2]Faculty of Geosciences and Environmental Engineering, Southwest Jiaotong University, Chengdu 610031, China
[3]College of Geography and Environmental Sciences, Zhejiang Normal University, Jinhua, 321004, China
[4]State Key Laboratory of Remote Sensing Science, Aerospace Information Research Institute, Chinese Academy of Sciences
[5]University of Chinese Academy of Sciences, Beijing 100049, China

*Correspondence to*: Xiaodan Wu (wuxd@my.swjtu.edu.cn)

**Abstract.** *In situ* measurements from sparsely distributed networks worldwide are a critical source of reference data for validating or correcting biases in satellite products. However, due to the substantial difference in spatial scales between *in situ* and satellite measurements, the two cannot be compared except for the fact that the underlying surface of in situ sites is
absolutely homogeneous. Instead, the *in situ* measurements needed to be upscaled to be matched with the satellite pixels. Based on the upscaling model we proposed as well as the consideration that *in situ* observation generally lacks spatial representativeness due to the widely distributed spatial heterogeneity, we have developed a coarse pixel scale ground "truth" dataset based on ground measurements of 416 *in situ* sites from the sparsely distributed observation networks. Furthermore, we thoroughly assessed the effectiveness of the dataset at sites with different degrees of spatial representativeness. The
results demonstrate that using this dataset in validation outperforms the direct comparison between satellite and *in situ* site measurements over heterogeneous surfaces when *in situ* measurement footprints are less than satellite pixel size. The accuracy of the reference data employed for validation or bias correction can be boosted by 17.09% over the regions with strong spatial heterogeneity. However, the degree of improvement with this dataset displays a decreasing trend as the reduction of spatial heterogeneity. At a global scale, the pixel scale ground "truth" dataset enhances the accuracy of pixel
scale reference data in general, with the overall RRMSE decreased by 6.04% compared to *in situ* single site measurements. Our results suggest that *in situ* single site measurements are limited in their ability to capture surface spatial variability information at a coarse pixel scale (i.e., the kilometer scale). The dataset we provided, which merges temporal information from ground-based observations and spatial information from high-resolution data, represents a valuable resource for validating and correcting worldwide surface albedo products over heterogeneous surfaces. To the best of our knowledge, this
dataset is unique in providing coarse pixel scale ground "truth" with the widest spatial distribution and longest time series.

## 1 Introduction

Surface albedo is an important variable in climate and biogeochemical models because it determines the amount of energy absorbed by the earth's surface. Coarse pixel (i.e., with a km pixel scale) satellite albedo products such as MODIS and NPP VIIRS have been widely used to tackle global challenges and support a range of initiatives (e.g., The Paris Agreement and Sustainable Development Goals). However, satellite albedo products generally suffer from different degrees of errors due to the error of satellite observation data and the limitation of the inversion algorithm, and the error of remote sensing product brings great uncertainty to the next application of the product. Taking albedo as an example, the change of albedo of 11% will cause a fluctuation of surface net radiation of 3.5 Wm$^{-2}$ on global and annual averages (GCOS-154, 2011), which in turn will cause the change in global temperature of 0.1K. An increase of $0.00106 \pm 0.00008$ (mean $\pm$ standard deviation) of albedo will cause the radiation at the top of the atmosphere to cool by $-0.15 \pm 0.1$ Wm$^{-2}$ (Ghimire et al., 2014). Therefore, it is very important to evaluate the uncertainty of remote sensing albedo products. In particular, when the error is relatively large, it is urgent to correct the error of remote sensing albedo products to improve the application accuracy of remote sensing products. Both the validation and correction of remote sensing products rely on reference data, which can represent the ground truth on the coarse pixel scale.

The sparsely distributed *in situ* sites (i.e., at most one site within a specific product grid cell) from the networks such as FLUXNET, BSRN, and SURFRAD provide an important data source for the validation of remote sensing albedo products (Chu et al., 2021; Augustine et al., 2000; Driemel et al., 2018). However, *in situ* measurements cannot be directly used as the coarse pixel scale truth if the footprint of *in situ* sites (depending on tower height) is far less than the scale of a coarse pixel. A practical method of using *in situ* site measurements as the coarse pixel scale truth is to conduct the spatial representativeness assessment of *in situ* sites (Román et al., 2009; Wang et al., 2014b; Moustafa et al., 2017). However, since these *in situ* sites were not originally established for the validation or bias correction of satellite products, only a small part of them was proved to be spatially representative, and most of them were rejected. Even for the representative site, the representativeness errors of *in situ* measurements are still inevitable, because land surface are not absolutely homogeneous throughout the year (Colliander et al., 2017; Xu et al., 2018; Lei et al., 2018; Williamson et al., 2018). Consequently, the representative *in situ* measurements are only limited to a few locations on the globe and cover discrete time periods, which cannot support a comprehensive validation and bias correction over a wide range of conditions (Loew et al., 2016).

To overcome the representative errors of *in situ* measurements and promote utilization ratio of *in situ* sites from these sparse networks in validation, Wu et al. (2020) have proposed an upscaling method specified for the single site *in situ* measurements. However, the effectiveness of this method has not been comprehensively assessed and its transferability to *in situ* sites all over the world is still unknown. As the continuation and deepening of our previous work (Wu et al., 2020), this study puts emphasis on the comprehensive evaluation and extensive use of this upscaling method based on 416 *in situ* sites throughout the world. Furthermore, a coarse pixel scale ground "truth" dataset was provided for validation and bias correction of satellite surface albedo products. The potential usage of this dataset was also discussed.

It is important to note that the Copernicus Global Terrestrial Monitoring Service partners have instituted a centralized validation database known as the Copernicus Global Terrestrial Product Validation Ground-based Observation Dataset (GBOV, http://gbov.copernicus.acri.fr), providing direct access to the set of reference measurements. However, the Copernicus GBOV ground-based observation dataset merely comprises 20 stations that provide albedo reference data, and the scope of these reference data is inadequate to systematically evaluate remote sensing products globally. Thus, our collection of ground-based "truth", which covers the widest spatial range and the longest time series on the coarse pixel scales, is essential to supplement the scientific efforts on existing albedo datasets and deliver a more precise and consistent albedo reference dataset on the coarse pixel scale for heterogeneous regions.

## 2 The experimental data

### 2.1 *In situ* site observation

In this study, *in situ* radiometric measurements from Surface Radiation (SURFRAD), Baseline Surface Radiation Network (BSRN), FLUXNET, Heihe Watershed Allied Telemetry Experimental Research (HiwaterWSN) (Li et al., 2013), and Huailai station (Ma et al., 2013), were incorporated to generate the coarse pixel scale "truth" dataset. These measurements include half-hourly observations of ecosystem fluxes and meteorological data. Figure 1 illustrates the spatial distribution of these *in situ* sites. The geographical distribution of these sites is predominantly concentrated in Europe and North America, accounting for 50 and 272 sites, which represent 12% and 65% of the total, respectively. These regions have a long history of conducting continuous and high-quality ecosystem flux measurements (Baldocchi et al., 2001). Additionally, there are several long-term observation sites located in tropical Amazonia, East Asia, and Australia. However, the coverage in Africa and polar regions remains limited both in terms of the number of sites and years observed. Despite this uneven geographical distribution, the selected *in situ* measurements ensure comprehensive coverage of the main plant functional types, including grasslands (GRA), croplands (CRO), woody savannas (WSA), deciduous broadleaf forests (DBF), mixed forests (MF), and evergreen needleleaf forests (ENF). These functional types are prominently represented, comprising 25%, 19%, 11%, 9%, 8%, and 6% of the sites, respectively.

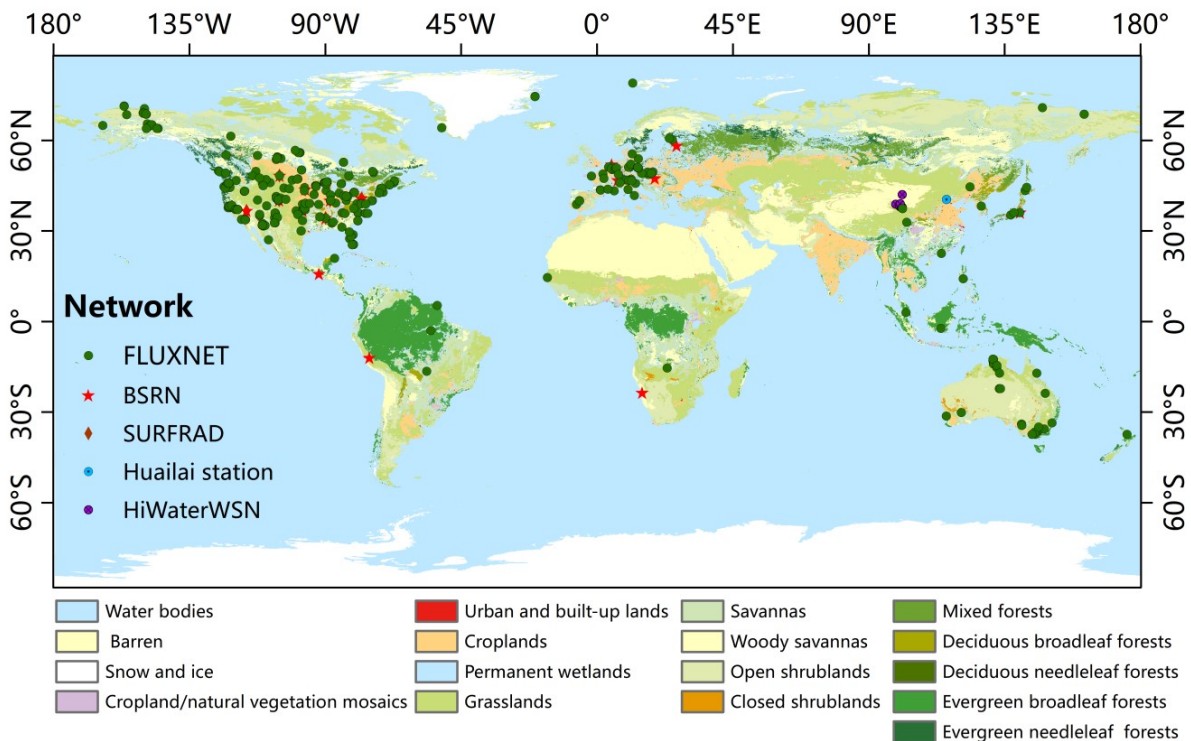

Figure 1: The distribution of the 416 *in situ* sites over different land cover types.

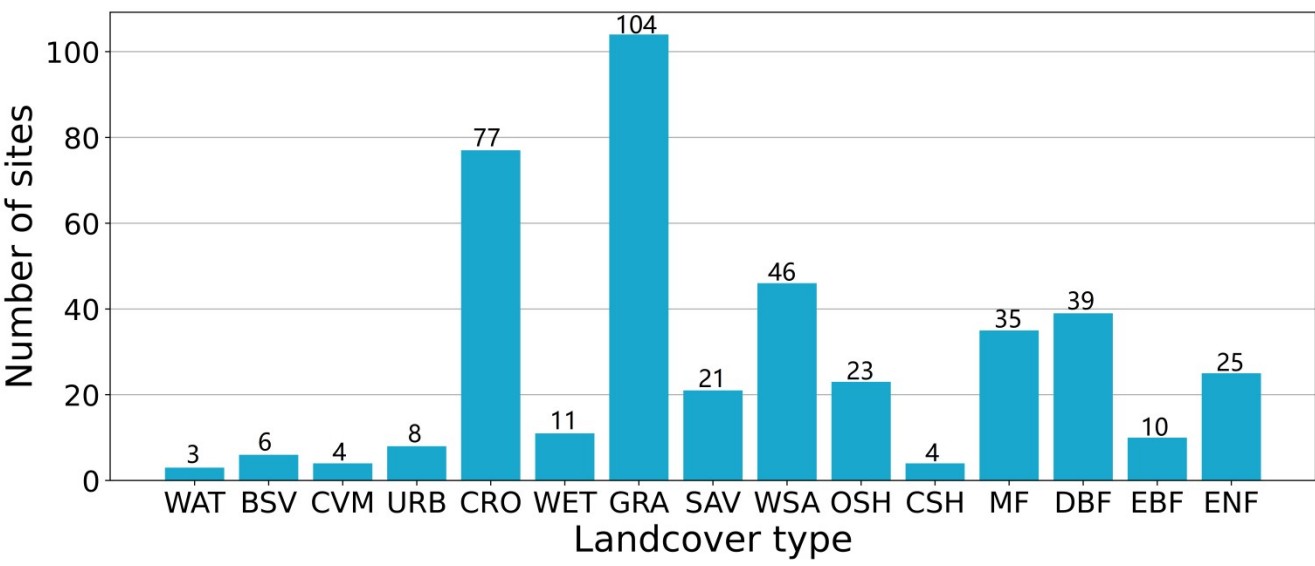

Figure 2: The distribution of stations across fifteen Land cover types indicated by International Geosphere-Biosphere Programme (IGBP). WAT, BSV, SNO, CVM, URB, CRO, WET, GRA, SAV, WSA, OSH, CSH, MF, DBF, EBF, and ENF are the abbreviations of water bodies, Barren, snow and ice, cropland/natural vegetation mosaics, urban and built-up lands, cropland permanent wetlands, grassland, savanna, woody savanna, open shrubland, closed shrublands, mixed forest, deciduous broadleaf

**forest, evergreen broadleaf forest, and evergreen needleleaf forests. The numeric value displayed above each bar in the chart indicates the total number of stations associated with the corresponding land cover type.**

These *in situ* sites were equipped with double pyranometers mounted back-to-back, one pointed downward and the other upward, measuring the downward radiation and upward radiation. Surface albedo is typically measured in the spectral range 280-2800 nm (which accounts for over 98.5% of surface solar radiation according to ASTM G-173 reference spectra), which is parallel to the broadband MODIS albedo (300-5000 nm). The slight inconsistency of spectral range between *in situ* and satellite measurements is negligible because the downwelling solar energy in the spectral region below 0.35 μm or above 2.2 μm is very little (Wright et al., 2014). It should be noted that the footprint of *in situ* sites is variable due to the different measurement heights of the albedometers from the underlying surface, which typically depends on both the height of albedometers and the height of the canopy top. Furthermore, the field of view of the various sensors and the ideal and non-ideal cosine response of the sensors need to be considered ( Balzarolo et al., 2011; Cescatti et al., 2012; Song et al., 2019; Marion, 2021). To reduce the possible effects of unstable lighting on flux measurements and align with satellite albedo products that generally report local solar noon albedo, *in situ* site measured blue-sky albedo was calculated using the ratio of the mean upward radiation to the mean downward radiation around local solar noon (11:00–13:00) as suggested by Lin et al. (2022).

### 2.2 The high-resolution albedo

Landsat Enhanced Thematic Mapper Plus (ETM+) imagery with eight distinct bands, including three in the visible spectrum, four in the infrared spectrum, and one panchromatic band, was incorporated as the pivotal component to upscale ground measurements to the scale of coarse pixel. This integration of ETM+ imagery served a twofold purpose, Firstly, it enabled the capture of spatial variability attributes at the coarse pixel scale surrounding each site, establishing the foundation for a robust upscaling model. Secondly, it facilitated coarse pixel-level aggregation within a 17 pixel × 17 pixel window (an approximately 510 m ×510 m area, considered as a coarse scale pixel), serving to be the reference value of the coarse pixel albedo. All Landsat ETM+ surface reflectance (SR) imagery from 2012-2018 were collected through Google Earth Engine (GEE) platform. There are an average of 65 ETM+ images collected at each site. Each ETM+ imagery has been atmospherically and geometrically corrected (Teixeira et al., 2020). The bad pixels such as those associated with cloud, cloud shadow, and saturated pixels, were identified and masked by the CFMask algorithm (Zhu et al., 2015; Zhu and Woodcock, 2012).

The retrieval of high-resolution surface albedo was executed using the algorithm proposed by Liang (2001). It relies on a spectral reflectance library and simulations conducted under various atmospheric and surface conditions with the Lambertian assumption so that the surface reflectance numerically equals spectral albedo (Liang et al., 2002). Notably, the anticipated accuracy of this algorithm approximates 0.02. The algorithm provides formulae for converting spectral information to broadband albedos for ETM+ imagery (Liang, 2001). In this study, we employed the following equation to calculate shortwave blue-sky albedo estimates.

$$\alpha_{short} = 0.356\alpha_1 + 0.130\alpha_3 + 0.373\alpha_4 + 0.085\alpha_5 + 0.072\alpha_7 - 0.0018 \qquad (1)$$

Where $\alpha_{short}$ denotes the shortwave blue-sky surface albedo, and $\alpha_i$ denotes the spectral albedo at the wavelength of $ith$ satellite spectral band.

### 2.3 Coarse pixel scale satellite albedo product

The MCD43A3 V061 product was used in this paper to serve as an example of coarse pixel satellite albedo products due to its wide acceptance. The shortwave (3000-5000 nm) albedo was extracted to match the spectral range of *in situ*-measured albedo. It provides black sky albedo (BSA) and white sky albedo (WSA) with a spatial resolution of 500 m and a temporal resolution of daily (Schaaf et al., 2002). The blue-sky albedo encompasses both direct and diffuse components, characterizing the albedo of the surface under actual atmospheric conditions. It can be expressed as a linear combination of BSA and WSA with an assumption of isotropic distribution of diffuse radiation. In this study, the following equation is used to calculate the MODIS blue-sky albedo (α) (Román et al., 2010; Lewis and Barnsley, 1994; Lucht et al., 2000; Pinty et al., 2005; Wang et al., 2019).

$$\alpha = \alpha_{WSA} \times r + \alpha_{BSA} \times (1 - r) \qquad (2)$$

where $r$ is the proportion of diffuse irradiation at a certain solar zenith angle (SZA), $\alpha$, $\alpha_{WSA}$ and $\alpha_{BSA}$ represent the blue-sky albedo, WSA, and BSA of MCD43A3, respectively. The proportion of diffuse radiation originates from light scattering and the reflection and transmission processes involving clouds and aerosols within a clear blue sky. In this study, we approximated the proportion of diffuse irradiation as a function of the cosine of the SZA at noon using an empirical statistical equation (i.e., Eq. (3)). Although this equation is approximate, it avoids the excessive amount of calculation while capturing the major phenomenon (Pinker and Laszlo, 1992). In fact, this empirical function has been widely used by previous studies (An et al., 2022; Mao et al., 2022; Wang et al., 2014a; Lewis and Barnsley, 1994).

$$r = 0.122 + 0.85 \times exp(-4.8 \times cos\,\theta) \qquad (3)$$

where $\theta$ denotes the solar zenith angle at local solar noon.

### 2.4 The ancillary dataset

Auxiliary data were used to detect the potential control factors that influence the accuracy of coarse pixel scale ground "truth". In this study, several common surface parameters, including elevation, land cover type, and spatial heterogeneity, were considered, as they were believed to be related to albedo. Elevation data were obtained from the Multi-Error-Removed Improved-Terrain (MERIT) Digital Elevation Model (DEM) (Mcclean et al., 2020; Yamazaki et al., 2017), with a high horizontal resolution of three arc-seconds (approximately 90 meters). MERIT DEM addresses a range of error components within the SRTM3 DEM, including stripe noise arising from sensor errors, speckle noise associated with surface reflectance, and absolute bias stemming from limited ground control points (Uuemaa et al., 2020). Land cover information was sourced

from MCD12Q1, with a spatial resolution of 500 m in an annual time step from 2012 to 2018. The IGBP classification scheme was selected in this study based on its recognized precision and widespread acceptance. With the consideration of the slight difference in land cover types at certain sites across different years, we opted to select the dominant land cover
160  type for each site.

## 3 Methodology

### 3.1 The upscaling model specified for single *in situ* site measurements

*In situ* measurements taken at a single *in situ* site can provide accurate measurements on the point scale and offer continuous temporal variation information for long time series. However, they are insufficient to represent albedo at the coarse-pixel
165  scale due to the spatial heterogeneity within the coarse pixel. High-resolution albedo maps can capture the spatial variation information within the coarse pixel. The basic idea of the upscaling model is to derive the upscaling coefficients based on high-resolution albedo maps and then apply these upscaling coefficients to long-term *in situ* measurements (Wu et al., 2020). In this way, both the spatial variation information and the temporal variation information of surface albedo can be captured through the combination of high-resolution albedo maps and long time series *in situ* measurements, deriving the long time
170  series pixel scale ground "truth" data.

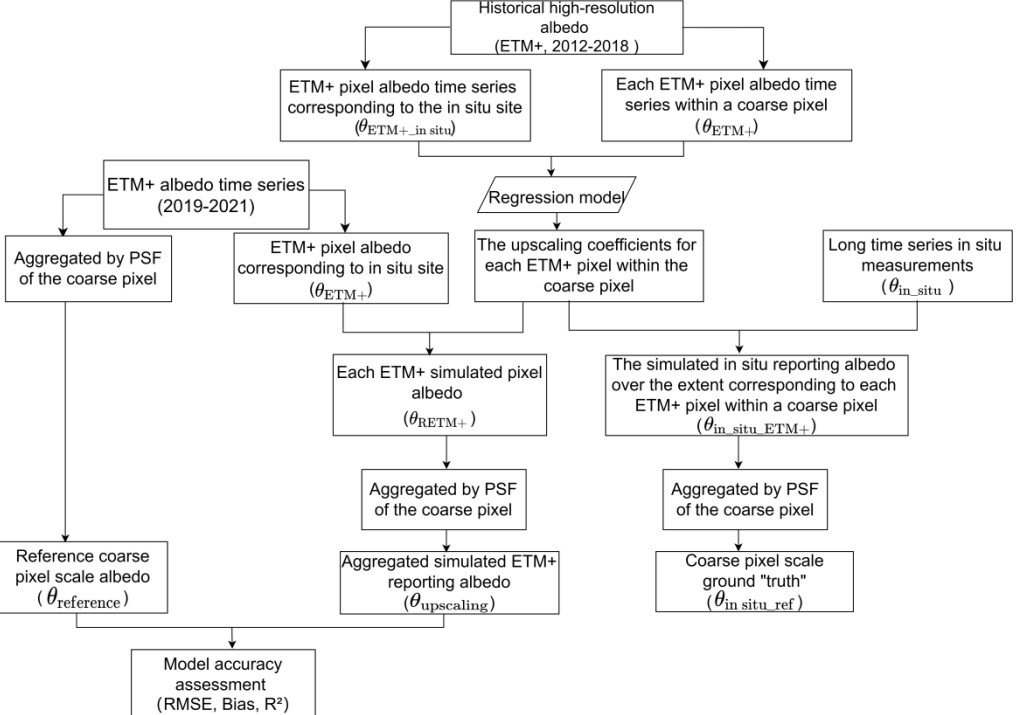

**Figure 3: The flowchart of generating coarse pixel scale ground "truth" based on upscaling model.**

Since the high-resolution albedo maps serve as an important linkage between *in situ* measurement scale and satellite coarse pixel scale, they should meet several requirements. First, its spatial resolution should be minimal to ensure the surface homogeneity within the fine pixel scale and stable radiation acquisition. Second, approximately 80% of the energy of the *in situ* observed signal originates from within 10-20 meters of the flux tower (Cescatti et al. 2012; Wang et al., 2014a), and the pixel size of high-resolution dataset should be near the footprint of *in situ* sites. Third, since the upscaling coefficients were determined by long time series high-resolution albedo maps and then were applied to long time series *in situ* measurements, the high-resolution albedo maps should cover at least one full cycle period, typically a year, to account for seasonal changes in surface heterogeneity caused by phenology and to guarantee the stability of the upscaling coefficients. For these reasons, the Landsat ETM+ albedo data were adopted in this study. In this paper, the Landsat ETM+ pixel is also called as subpixel relative to the coarse pixel.

The upscaling coefficients were calculated for each subpixel within the coarse pixel extent by establishing a regression relationship between one subpixel albedo time series and the subpixel albedo time series corresponding to the *in situ* site (Eq. (4)). To avoid the uncertainty caused by different data sources, both of them were simulated by ETM+ albedo. Using the same data source can reduce the influence of errors of ETM+ albedo to a certain extent.

$$\theta_{ETM+}(x, y, d) = W(x, y)^T \theta_{ETM+\_in\ situ'}(d) \tag{4}$$

and

$$\theta_{ETM+\_in\ situ'}(d) = [1, \theta_{ETM+\_in\ situ}(d)]^T \tag{5}$$

where x and y correspond to the location of a single ETM+ pixel within a coarse pixel, while d denotes the date of the ETM+ albedo map. $\theta_{ETM+\_in\ situ}$ denotes the ETM+ albedo time series corresponding to the *in situ* site. The vector W represents the upscaling coefficients to be derived.

To ensure a robust estimation, a cost function J is established by combining all the ETM+ albedo data throughout the whole time series (i.e., 2012-2018).

$$J = \min\{\sum_{d=1}^{L}[\theta_{ETM+}(x, y, d) - W(x, y)^T \theta_{ETM+\_in\ situ'}(d)]^2\} \tag{6}$$

Using the ordinary least-squares (OLS) algorithm, the vector of coefficients W can be obtained by minimizing the cost function.

$$W(x, y) = [\theta_{ETM+\_in\ situ'}^T \theta_{ETM+\_in\ situ'}]^{-1} \theta_{ETM+\_in\ situ'}^T \theta_{ETM+}(x, y) \tag{7}$$

When the upscaling coefficients were determined, they were applied to *in situ* site measurements ($\theta_{in\ situ}$) to simulate the *in situ* reporting of surface albedo ($\theta_{in\ situ\_ETM+}$) over each ETM+ pixel within the coarse pixel:

$$\theta_{in\ situ\_ETM+}(x, y, d) = W(x, y)^T \theta_{in\ situ}(d) \tag{8}$$

Then the coarse pixel ground "truth" ($\theta_{in\ situ\_ref}$) can be derived by aggregating all the $\theta_{in\ situ\_ETM+}$ within the coarse pixel using the point spread function (PSF) of the MODIS albedo characterized by Peng et al. (2015).

$$\theta_{in\ situ\_ref}(d) = \frac{\int_{(x,y)\in D} f_{PSF}(x,y)\theta_{in\ situ\_ETM+}(x,y,d)}{\int_{(x,y)\in D} f_{PSF}(x,y)} \tag{9}$$

where $D$ denotes the spatial extent of the coarse pixel, and $f_{PSF}$ represents the PSF.

The upscaling coefficients in Eqs. (8-9) remain time-independent and can be used to upscale *in situ* measurements throughout the entire time series, except when sudden changes like wildfires or deforestation on the land surfaces occur. It is noteworthy that a high-resolution albedo map is no longer a prerequisite for the practical upscaling process once the upscaling coefficients have been obtained.

### 3.2 The evaluation of upscaling models and pixel scale ground "truth"

### 3.2.1 The evaluation of the upscaling model

The accuracy of the generated albedo dataset hinges significantly on the quality of the upscaling model employed. In the study, a critical component of the upscaling approach involves the acquisition of upscaling coefficients derived from 30-meter ETM+ albedo covering the period from 2012 to 2018. Consequently, the accuracy of the upscaling model is intricately tied to the performance of these coefficients. Inspired by the evaluation approach proposed by Wu et al. (2016), we conducted an assessment to gauge the accuracy and robustness of these upscaling coefficients. This evaluation involved a comparison between aggregated fine scale albedo and reference coarse scale albedo, utilizing ETM+ albedo data acquired from 2019 to 2021. The aggregated fine scale albedo with the upscaling coefficients can be determined with the Eqs. (10-11), the reference coarse scale albedo is the aggregated ETM+ albedo on coarse pixel scale as recommended by Wu et al. (2016) (Eq(12)).

$$\theta_{RETM+}(x,y,d) = W(x,y)^T \theta_{ETM+}(d) \tag{10}$$

$$\theta_{upscaling}(d) = \frac{\int_{(x,y)\in D} f_{PSF}(x,y)\theta_{RETM+}(x,y,d)}{\int_{(x,y)\in D} f_{PSF}(x,y)} \tag{11}$$

$$\theta_{reference}(d) = \frac{\int_{(x,y)\in D} f_{PSF}(x,y)\theta_{ETM+}(d)}{\int_{(x,y)\in D} f_{PSF}(x,y)} \tag{12}$$

Where $\theta_{ETM+}$, $\theta_{RETM+}$ are ETM+ albedo corresponding to *in situ* site and each ETM+ simulated pixel albedo incorporating upscaling coefficients and $\theta_{ETM+}$. $\theta_{upscaling}$ denotes the upscaling results based on the $\theta_{ETM+}$ and upscaling coefficients. $\theta_{reference}$ represents the reference coarse pixel scale albedo.

The similarity and consistency between the $\theta_{upscaling}$ and $\theta_{reference}$ were evaluated by three metrics: bias, coefficient of determination ($R^2$), and root-mean-square error (RMSE).

$$RMSE = \sqrt{\sum_{d=1}^{L}(\theta_{upscaling}(d) - \theta_{reference}(d))^2 / L} \tag{13}$$

$$Bias = \sum_{d=1}^{L}(\theta_{upscaling}(d) - \theta_{reference}(d))/L \tag{14}$$

$$R^2 = \frac{[\sum_{d=1}^{L}(\theta_{upscaling}(d) - \overline{\theta_{upscaling}})(\theta_{reference}(d) - \overline{\theta_{reference}})]^2}{\sum_{d=1}^{L}(\theta_{upscaling}(d) - \overline{\theta_{upscaling}})^2 \sum_{d=1}^{l}(\theta_{reference}(d) - \overline{\theta_{reference}})^2} \tag{15}$$

### 3.2.2 Assessment of coarse pixel scale ground "truth"

When the upscaling coefficients were determined, they can be applied to *in situ* measurements to derive pixel scale ground "truth" that aligns with the spatial resolution of coarse resolution products. The evaluation process adheres to the previously outlined methodology (Section 3.2.1). Namely, the reference coarse pixel scale albedo ($\theta_{reference}$) was also utilized to assess the accuracy of coarse pixel scale ground "truth" ($\theta_{in\ situ\_ref}$) as suggested by previous studies (Wu et al., 2016; Wu et al., 2020) as follows:

$$RMSE = \sqrt{\sum_{d=1}^{L}(\theta_{in\ situ\_ref}(d) - \theta_{reference}(d))^2 / L} \tag{16}$$

To eliminate the influence of the magnitude of surface albedo on accuracy indicators, the relative root-mean-square error (RRMSE) was used here, which is defined as the ratio of RMSE to the mean surface albedo at the coarse pixel scale.

$$RRMSE = \frac{RMSE}{\overline{\theta_{in\ situ\_ref}}} \times 100\% \tag{17}$$

where $\theta_{in\ situ\_ref}$, $\overline{\theta_{in\ situ\_ref}}$ represents coarse pixel scale "truth" and the mean value of coarse pixel scale "truth", and L denotes the length of the temporal sequence of data.

It is important to note that the $\theta_{reference}$ dataset used here is not necessarily the same as that in Section 3.2.1 due to the different match pairs in the temporal domain. In Section 3.2.1, the $\theta_{reference}$ is the result of the match between $\theta_{ETM+}$ and $\theta_{RETM+}$ as shown in Eqs. (11-12). By contrast, in this section, the $\theta_{reference}$ is the result of the match between $\theta_{ETM+}$ and $\theta_{in\ situ}$ as shown in Eqs. (8-9) and Eq. (12). In addition to these accuracy indicators, the performance of coarse pixel scale ground "truth" was also assessed through the comparison with single *in situ* site measurements.

### 3.2.3 Measure of Spatial Heterogeneity

Spatial heterogeneity is a critical factor influencing the spatial scale match between *in situ* and satellite measurements, because it reduces the spatial representativeness of *in situ* measurements (Wu et al., 2022). It refers to the uneven distribution of surface albedo within a coarse pixel. A pixel that exhibits spatial heterogeneity denotes that the value of surface albedo at one location is different from that of other locations. To quantify the spatial heterogeneity of surface albedo within a coarse

pixel, the spatial variability (standard deviation, Std) of all subpixel albedos within a coarse pixel was calculated as recommended by previous studies (Colliander et al., 2017; Jin et al., 2003). Here, the subpixel albedos denote the high-resolution pixel albedo (i.e., Landsat ETM+ pixel albedo) within the coarse pixel.

$$Std = \sqrt{\frac{1}{L-1}\sum_{i=1}^{L}(Z_i - \overline{Z})^2} \tag{18}$$

where $Z_i$ denotes the high-resolution albedo and $\overline{Z}$ is the averaged albedo of all high-resolution albedos within the extent of the coarse pixel. L refers to the number of high-resolution albedo pixels with a coarse pixel.

## 4 Results and Discussion

### 4.1 The performance of the upscaling model

The performance of the upscaling coefficients has been comprehensively evaluated over the 416 *in situ* sites. The wide spread of *in situ* sites across different elevations, different land cover types, and different degrees of spatial heterogeneity can ensure the objectivity of the evaluation results. To show the agreement between $\theta_{upscaling}$ and $\theta_{reference}$ more intuitively, we present the scatterplots between them in Fig. 4. As shown in Fig. 4, the scatterplots between $\theta_{upscaling}$ and $\theta_{reference}$ are generally distributed around the 1:1 line, with $R^2$ close to 0.9. The upscaling coefficients show no systematic error, indicated by the biases close to 0. However, the performance of the upscaling models is site-dependent. For instance, the accuracy of the upscaling models over the US-Ha2 site obviously outperforms that of the IT-Tor site (Fig. 4(c vs. f)). It is apparent that the RMSE, $R^2$, and the sample size are unrelated. For instance, US-UMB and CA-NS2 share comparable environmental conditions, despite their dissimilar sample sizes, yet their RMSE and $R^2$ are similar. To fully understand the performance of upscaling coefficients in different conditions, the accuracy indicators of the upscaling coefficients throughout these 416 *in situ* sites were summarized as the histograms (Fig. 5).

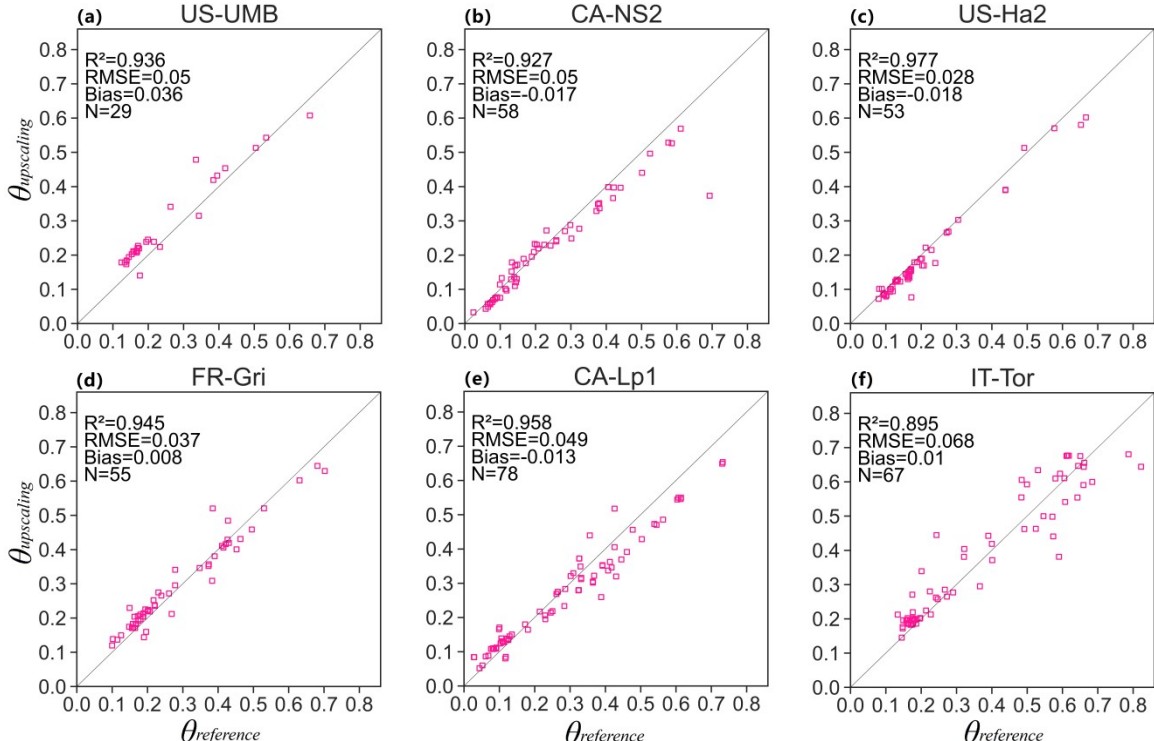

**Figure 4: The scatter plots between the upscaling results ($\theta_{upscaling}$) with the upscaling models and the coarse pixel scale reference ($\theta_{reference}$). Only parts of the results are shown for conciseness. Specifically, only one *in situ* site is shown for each land cover type.**

**Table 1: Description of the *in situ* sites used in the model performance analysis.**

| Networks | US-UMB | CA-NS2 | US-Ha2 | FR-Gri | CA-Lp1 | IT-Tor |
|---|---|---|---|---|---|---|
| Location (lon, lat) | (-84.7138, 45.5598) | (-98.5247, 55.9058) | (-72.1779, 42.5393) | (13.51259, 50.9500) | (-122.8414, 55.1119) | (7.5781, 45.8444) |
| Spatial heterogeneity | 0.0133079 | 0.0640852 | 0.0065224 | 0.5564959 | 0.18694994 | 1.01929451 |
| Elevation (m) | 236.72682 | 271.09771 | 367.29669 | 377.65914 | 749.265564 | 2162.78979 |
| Land cover type | DBF | EBF | MF | CRO | WSA | GRA |

Based on the results presented in Fig. 5, it can be seen that the overall accuracy of the upscaling coefficients is satisfactory. The biases range from -0.06 to 0.10, and more than 90% of them are within the range of ± 0.02 (Fig. 5(b)). The highest density of $R^2$ is between 0.9 and 1 as shown in Fig. 5(c), and only a small part of the sites show a relatively small $R^2$ of lower than 0.8 but larger than 0.5. Nevertheless, it should be noted that those sites exhibit a more scattered distribution of RMSE values, with a maximum of 0.1 and a minimum of 0.01 (Fig. 5(a)). The highest density is between 0.03 and 0.05 for RMSE.

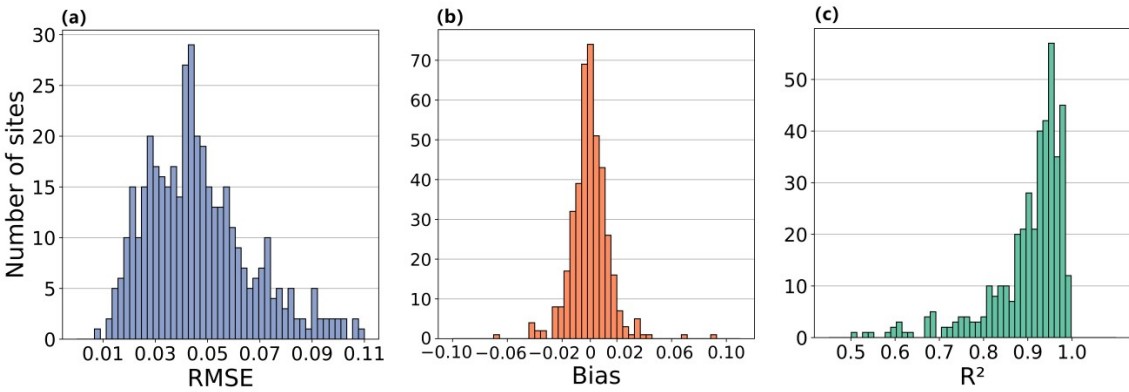

**Figure 5: Distribution of RMSE (a), Bias (b), and R² (c) of the upscaling coefficients. The histograms presented here combine the results of the 416 _in situ_ sites.**

As can be seen from the accuracy distribution of the upscaling model, the proportion of sites with reasonable RMSE and $R^2$ is more than 65%. Moreover, the RMSE and $R^2$ show consistent instructions about the performance of the upscaling model as shown in Fig. 6. For instance, in the case of optimal $R^2$, the RMSE is very likely to be less than 0.05. By contrast, the poor RMSE is generally accompanied by poor $R^2$ of the model. The distribution of the sites with poor RMSE and $R^2$ is dispersed and not location-specific. Both GCOS (GCOS-154, 2011) and CEOS LPV albedo best practice protocols (Wang et al., 2019) indicate the better performance of BSRN than other networks. However, this phonemenon does not occur with this upscaling model given the comparable RMSE and $R^2$ among different networks.

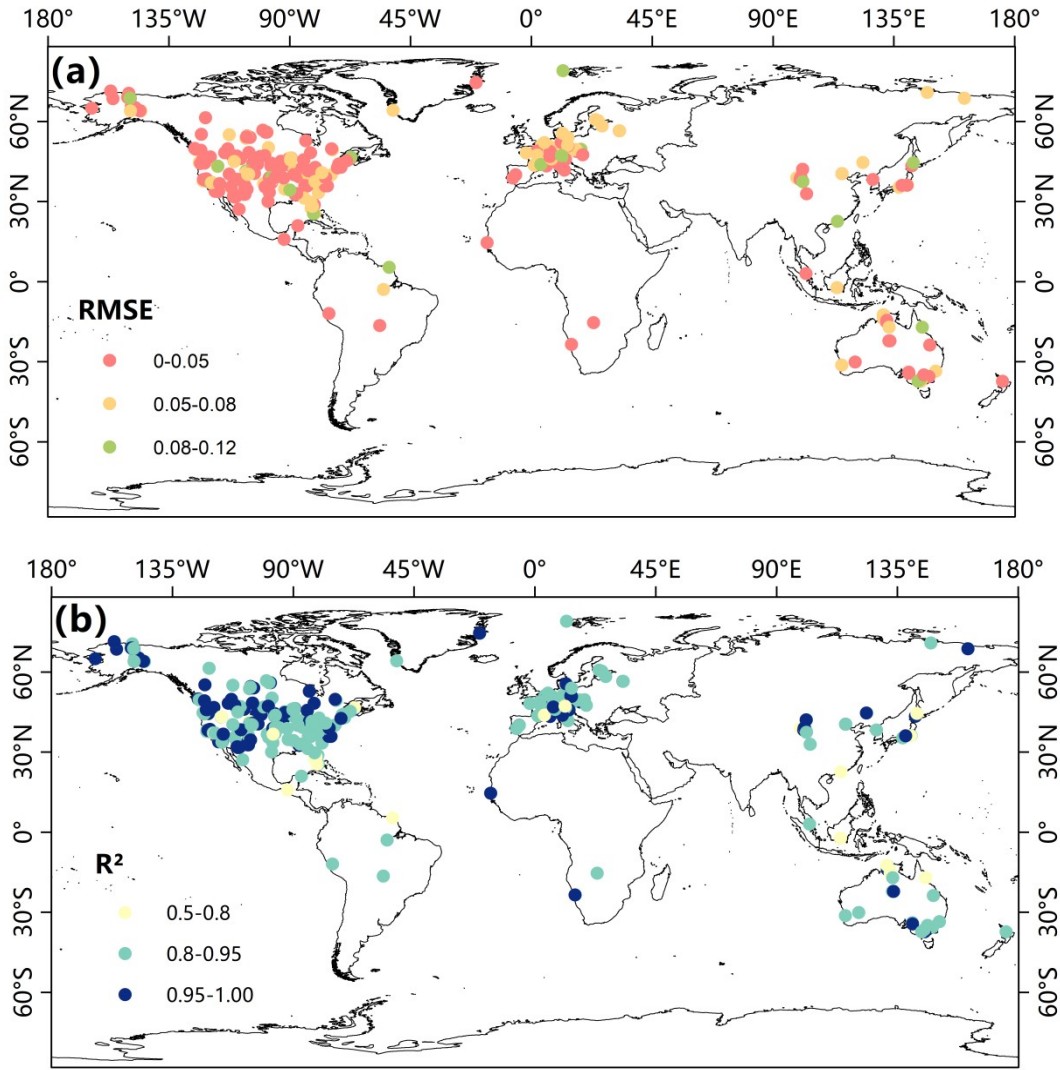

**Figure 6: Spatial distribution of RMSE (a) and R² (b) of the upscaling model.**

295    Given the fact that the accuracy of the upscaling models shows great variability, it is necessary to explore the influence factors on the performance of the upscaling models. In this study, the effect of land cover type, elevation, and spatial heterogeneity were considered. The influence of spatial heterogeneity on the accuracy of the upscaling model is displayed in Fig. 7. The RMSE exhibits a significant positive correlation with spatial heterogeneity (Fig. 7(a)), with superior performance often observed in areas with lower spatial heterogeneity. Similarly, the $R^2$ of different sites typically decreases with the

300    increase of spatial heterogeneity (Fig. 7(b)). It is worth noting that when the spatial heterogeneity exceeds 0.1, the $R^2$ of the model fluctuates considerably, indicated by the larger height of the boxplots. Based on these results, it can be seen that spatial heterogeneity has enormous implications for the performance of the upscaling models. One possible reason is that the

assumption of a linear relationship between the subpixel albedo of other locations and the subpixel albedo containing the *in situ* site cannot be satisfied over the surface with large spatial heterogeneity.

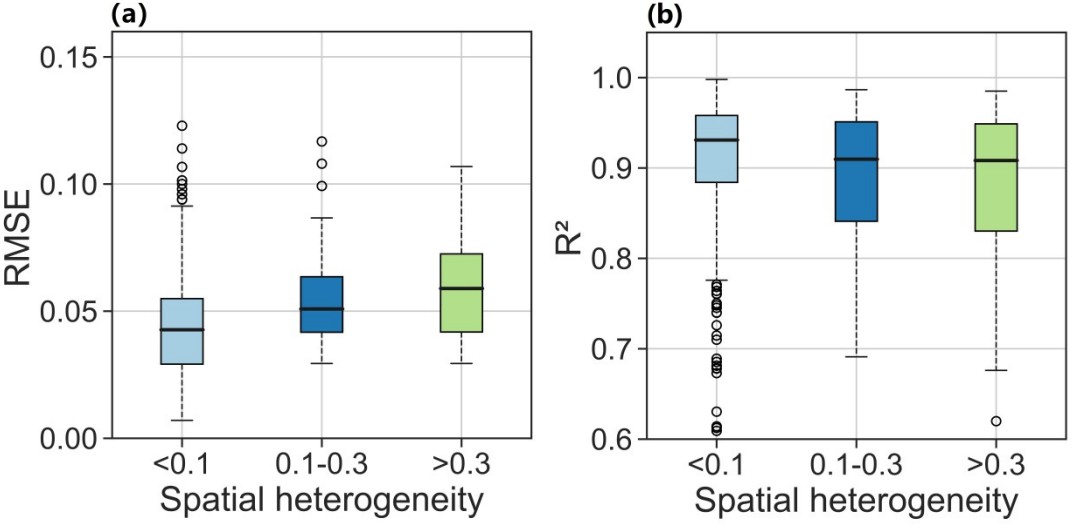

**Figure 7: Boxplots showing the dependence of RMSE (a) and R² (b) of the upscaled albedo on spatial heterogeneity. Three different degrees of spatial heterogeneity are marked by different colors. Black lines indicate median values. Outliers are values that are farther than 1.5 interquartile ranges. The number of *in situ* sites with spatial heterogeneity of [0-0.1], [0.1-0.3], and [0.3-1.5] are 337, 49, and 30, respectively.**

Figure 8 illustrates the RMSE and $R^2$ of the upscaling model as a function of elevation. Notably, the upscaling model exhibits the highest accuracy at elevations below sea level, with the lowest median RMSE of about 0.03 and the highest median $R^2$ of more than 0.95. In contrast, the model performs poorly at elevations exceeding 2500 m, with the highest RMSE and the lowest $R^2$. However, there are no significant trends of RMSE and $R^2$ in the areas with an altitude between 0 and 2500 m above sea level. There is merely a slight decrease trend as the altitude increases from 0-200 m to 500-1500 m, but then a slight increasing trend appears as the altitude increases from 500-1500 m to above 2500 m. Both RMSE and $R^2$ exhibit significant variability, indicated by the large heights of the boxplot, except for regions where the elevation is less than 0 m. These results imply that the accuracy of the upscaling models is subject to a diverse array of factors. The good performance of the upscaled model in the area below sea level may be attributed to the limited spatial variability given that the spatial heterogeneity of the region below sea level is less than 0.1 as shown in Fig. 9. By contrast, the poor accuracy of the upscaling model above 2500 m may be partly attributed to the fact that the areas above 2500 m have complex and undulating topography with elevations ranging from 2500-4000 m. These results demonstrate that the substantial variation in elevation also significantly impacts the performance of the upscaling model.

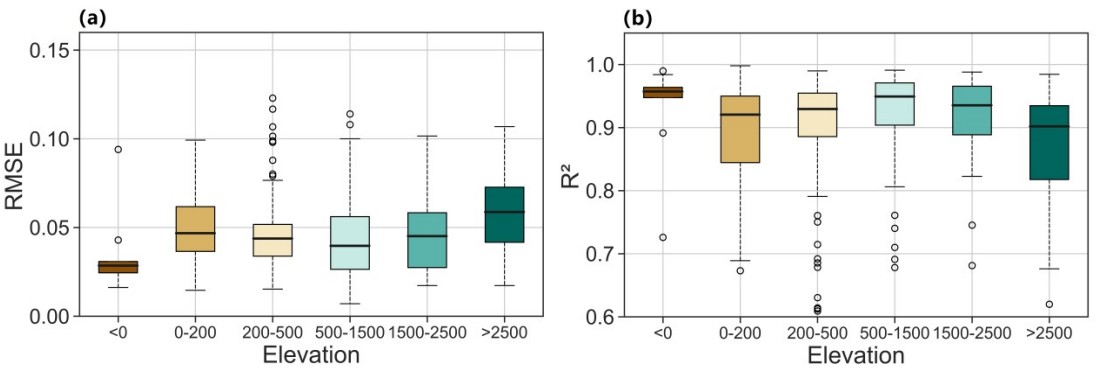

**Figure 8: The variations of RMSE (a) and R²(b) of with elevation. The accuracy of upscaling models responds to different elevation when the elevation is below sea level (n=9), [0-200] (n=114), [200-500] (n=162), [500-1500] (n=78), [1500-2500] (n=32), above 2500 m (n=21). Black lines indicate median values. Outliers are values that are farther than 1.5 interquartile ranges.**

As illustrated in Figure 9, the RMSE significantly rises with the augmentation of spatial heterogeneity at each level of elevation. This indicates that spatial heterogeneity plays a dominant role in determining the performance of the upscaling models. Nevertheless, the influence of spatial heterogeneity seems to be related to the elevation. As shown in Figure 9(a), there is a tendency for the difference in median RMSE between different levels of spatial heterogeneity, which increase gradually with the elevation, and a similar pattern is observed for $R^2$. However, the trends of RMSE and $R^2$ with altitude are not the same for each level of spatial heterogeneity. The trends for regions with low spatial heterogeneity (< 0.1) were not significant. In contrast, regions exhibiting high spatial heterogeneity (> 0.3) showed an increasing/decreasing trend for the RMSE/$R^2$ with elevation, particularly for the area above 500 m.

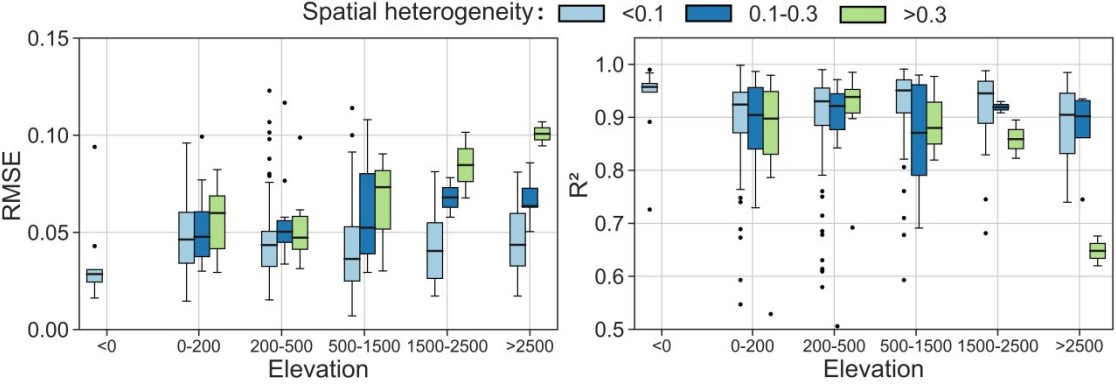

**Figure 9: The plots show combined results of the RMSE (a) and R²(b) variations based on elevation and spatial heterogeneity. Black lines indicate median values. Outliers are values that are farther than 1.5 interquartile ranges.**

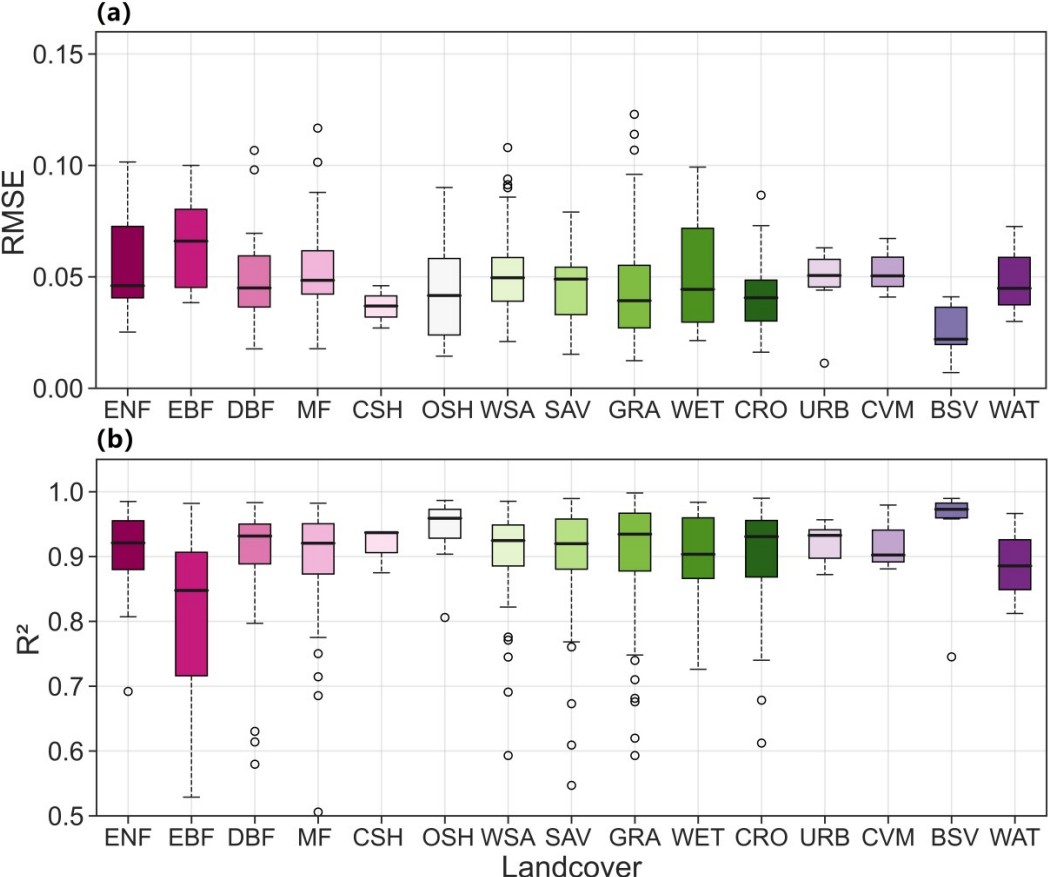

**Figure 10: The variations in RMSE (a) and R²(b) are dependent on landcover, different colors refer to the fifteen different land cover types. The accuracy of upscaling models responds to different land cover types when the land cover types are ENF (n=25), EBF (n=10), DBF (n=39), MF (n=35), CSH (n=4), OSH (n=23), WSA (n=46), SAV (n=21), GRA (n=104), WET (n=11), CRO (n=77), URB (n=8), CVM (n=4), BSV (n=6), WAT (n=3). Black lines indicate median values. Outliers are values that are farther than 1.5 interquartile ranges.**

The influence of land cover type on the accuracy of the upscaling model is displayed in Fig. 10. It is revealed that the performance of the model is considerably insufficient for EBF, as most of the RMSEs exceed 0.05 and the $R^2$ values below 0.90. By contrast, the model delivers the optimal outcomes for barren (BSV), with the smallest RMSE being approximately 0.01 and a relatively high $R^2$ value of around 0.97. The accuracy of the model is comparable across all other surface cover types, with RMSEs and $R^2$ values of approximately 0.05 and 0.90, respectively. Additionally, the accuracy remains consistent for CSH, URB, and CVM based on the small range of RMSE boxplots, indicating an overall stable model performance. On the other hand, the RMSE boxplots for ENF and EBF exhibited a significant range of values, highlighting the substantial variation in model performance in these regions. Furthermore, the impact of land cover type is elaborately

connected to the influence of spatial heterogeneity. Figure. 11 illustrates that there is limited spatial heterogeneity for the BSV, while considerable locations with EBF exhibit pronounced spatial heterogeneity.

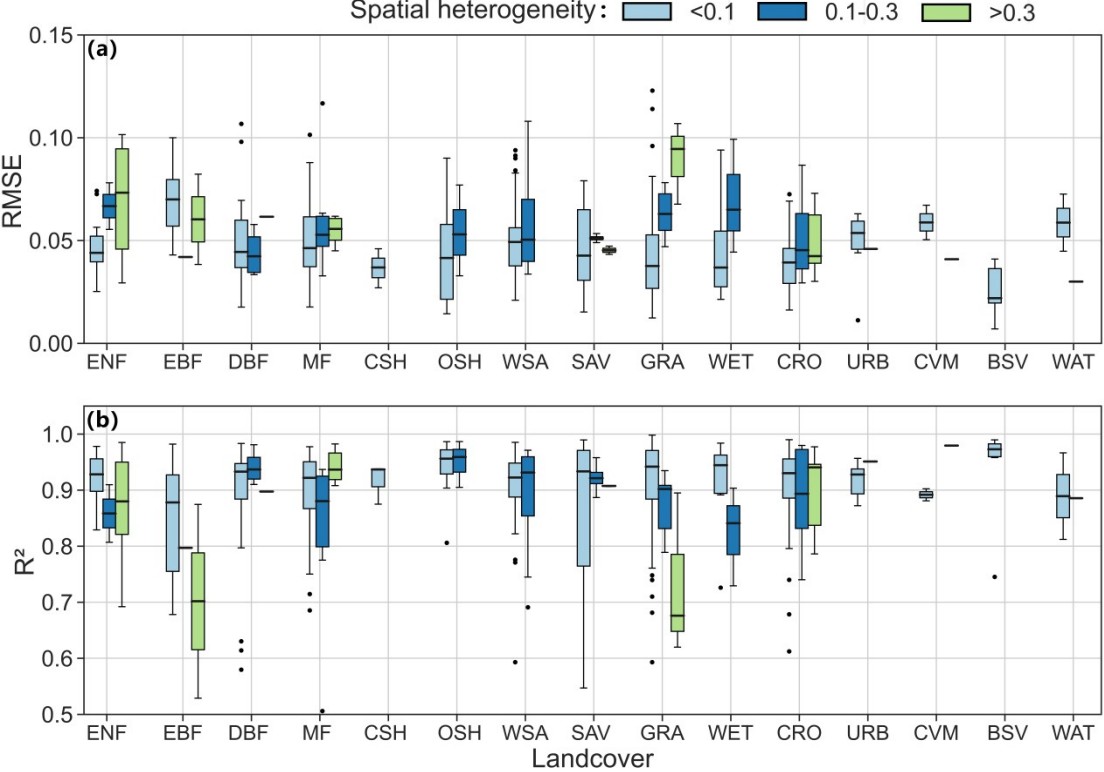

As shown in Fig. 11(a), the RMSE of the upscaling model basically presents an increasing trend with spatial heterogeneity over each land cover type, further indicating the dominant role of spatial heterogeneity in determining the accuracy of the upscaling model. The stable accuracy of the upscaling models at OSH, URB, and CVM is attributed to fewer sites with relatively large spatial heterogeneity. Nevertheless, the influence of spatial heterogeneity shows dependence on land cover type, which is most significant on GRA.

### 4.2 The accuracy of the pixel scale ground "truth"

Since there are a considerable number of *in situ* sites, the accuracy of pixel scale ground "truth" was summarized as the boxplots (Fig. 12). For comparison purposes, the errors of single site measurements when they were directly used as the pixel scale reference were also calculated and summarized as the boxplots. It can be seen that the errors of pixel scale ground "truth" show a slight variation with spatial heterogeneity, with the median RRMSE ranging from 88.35% to 113.69% and then to 138.26%, resulting in the overall RRMSE of 95.20%. It is important to note that this variation pattern is not the same

as the accuracy of the upscaling model, which shows a monotonous decrease trend with the increase of spatial heterogeneity. The wide range of the boxplots shows that the accuracy of the pixel scale ground "truth" is also influenced by other factors.

Although the errors of the pixel scale ground "truth" are not negligibly small, it is important to note that this kind of error cannot reveal the absolute accuracy of pixel scale ground "truth" given that the reference data itself contains errors. In fact, the focus of this evaluation is not the value of RRMSEs but the difference of RRMSEs between the pixel scale ground "truth" and single *in situ* site measurements. It can be seen that the accuracy of the pixel scale ground "truth" is consistently better than the single site measurements over the surfaces with different levels of spatial heterogeneity as shown in Fig. 12. The

smaller RRMSE of pixel scale ground "truth" (95.20% vs. 101.24%) indicates that this dataset can improve the overall accuracy of pixel scale reference data a lot compared to the single site measurements. Nevertheless, the degree of the improvement depends on the situation, which is the most significant over the sites with the strongest spatial heterogeneity, with the RRMSE decreasing from 155.35% to 138.26%. The *in situ* sites with medium spatial heterogeneity follow, with the RRMSE decreasing from 127.91% to 113.69%. The improvements are the smallest over the sites with the smallest spatial

heterogeneity, with the RRMSE decreasing from 92.03% to 88.35%. Hence, it can be concluded that the degree of improvements of this dataset shows an increasing trend with spatial heterogeneity. Furthermore, the accuracy of the pixel scale ground "truth" dataset is more stable than that of the single site measurements, indicated by the smaller height of the boxplots of the former.

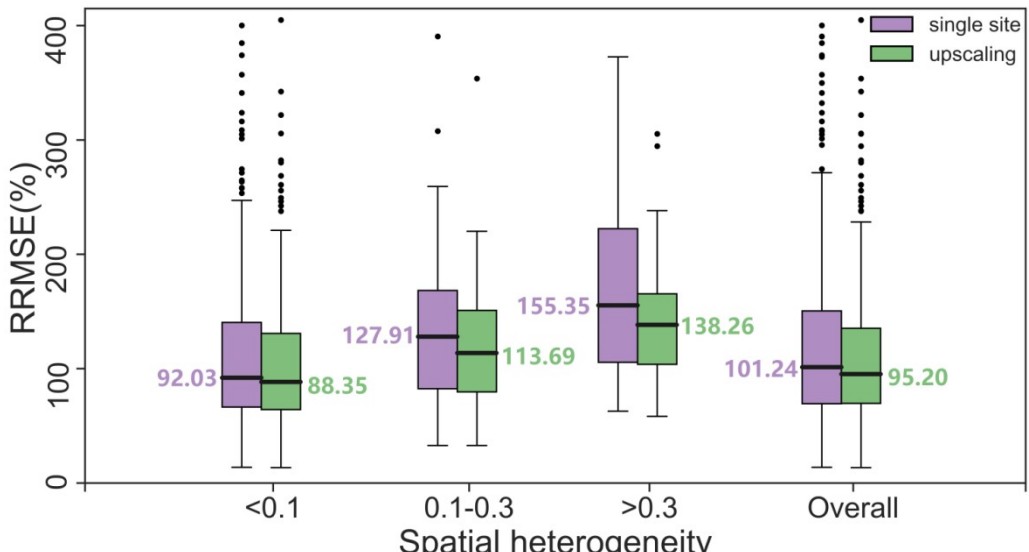

**Figure 12: The boxplots of RRMSE of pixel scale ground "truth" and single site measurements. The boxplots are categorized by different degrees of spatial heterogeneity and overall accuracy. The median of the boxplots is indicated by the numbers around the plots. Black lines indicate median values. Outliers are values that are farther than 1.5 interquartile ranges.**

The aforementioned results confirm the effectiveness of our pixel scale ground "truth" dataset in different scenarios, which is superior to single site measurements, whether for the sites with higher or lower spatial heterogeneity. The improvement of

this dataset is more significant over the heterogeneous sites. Hence, it is highly helpful over heterogeneous surfaces in the validation or bias correction of satellite albedo products.

## 4.3 The usage of the pixel scale ground "truth" dataset

Validation is important for both satellite product manufacturers and end users as it provides a quantitative assessment of the advantages and disadvantages of satellite products. However, due to diverse locations, coverage, scaling, and representation

of *in situ* measurements, the accuracy of satellite products can vary greatly. As a result, direct comparisons of validation results are challenging, which ultimately limits the overall usefulness of satellite products. As highlighted by GCOS (GCOS-200, 2016), one solution to tackle this issue is to adopt a comprehensive and uniform validation process that relies on a standardized, consistent, and systematized reference dataset. The ground "truth" for the pixel scale was acquired using a standardized operational procedure that leveraged a considerable number of measurements collected from *in situ* sites

scattered around the world. Such standardization enables the fair comparison between the accuracy of various satellite products of the same ECV. Thus, it provides a foundation for coordinating the use of diverse satellite albedo products and maximizing their potential capabilities. Fig. 13 presents an example of the validation of MCD43A3 V0061 using the coarse pixel scale ground "truth" dataset.

Apart from serving as the reference for evaluating the accuracy of satellite albedo products, the pixel scale ground "truth"

database can also used as a reference for addressing the influence of biases in satellite albedo products. Various models have been developed for such bias corrections (Wang et al., 2022). For this research, the CDF (cumulative distribution function) approach (Calheiros et al., 1987) has been employed to correct the bias in MCD43A3 V0061 as an example. As indicated in Figure 13 and Table 1, correcting the bias generally makes satellite albedo products more accurate, especially in regions with significant heterogeneity (Fig. 13(d-f)). Therefore, it is reasonable to assume that this dataset could enhance the quality of

satellite albedo products in regions with prominent surface heterogeneity.

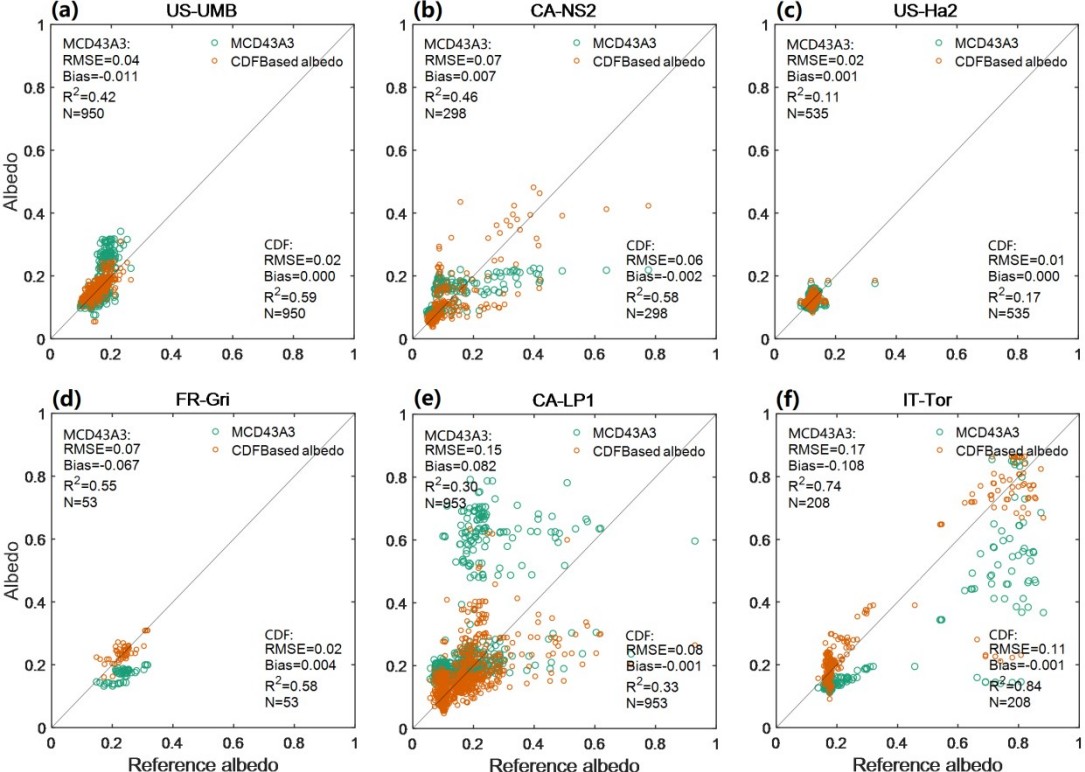

**Figure 13: The scatter plots between the MCD43A3 and pixel scale ground "truth" (green dots) as well as the scatterplots between CDF-corrected MCD43A3 and pixel scale ground "truth" (brown dots). US-UMB, CA-NS2, US-Ha2 represent the regions with small heterogeneity, FR-Gri, CA-LP1, IT-Tor represent the regions with medium and strong heterogeneity.**

**5 Data availability**

The processed coarse pixel scale ground "truth" dataset is publicly available and can be obtained through Zenodo (https://zenodo.org/record/8362185; Pan et al., 2023). The dataset files are available in the machine-readable data format (.tab) and have been categorized separately into folders for FLUXNET, BSRN, SURFRAD, HIWATERWSN, and Huailai Station, facilitating easy accessibility and utilization.

**6 Conclusion**

The validation and correction of satellite reflectance products is essential to promote the reasonable use of such products in various scientific applications. Typically, these tasks depend on extensive *in situ* single site-based albedo measurements. However, as satellite-based albedo and tower-based albedo are generated at different spatial scales, direct comparison can only be performed on certain homogeneous surfaces. Nevertheless, spatial heterogeneity is a fundamental feature of most

land surfaces, which limits the spatial representativeness of measurements from single sites. Therefore, the most critical aspect of validation and correction is obtaining ground truth albedo at the pixel level based on field measurements.

However, the methods used to acquire ground truth at the pixel scale differ greatly in regard to field measurements, location, coverage, scale, and representation, leading to various accuracy levels of pixel scale "truth" datasets. Consequently, most validation/correction outcomes are not easily comparable, thereby further hindering the applications of satellite products. There is a requirement for a consistent, impartial, and representative coarse pixel scale ground "truth" dataset of surface albedo. As far as is currently known, such a dataset with global coverage is currently lacking. Therefore, we have developed a coarse pixel scale ground "truth" dataset using data collected from 416 *in situ* sites in sparsely distributed observational networks including SURFRAD, BSRN, and Fluxnet, and a specified upscaling model for individual site measurements to fill this gap.

The suitability of the upscaling model for applying to the *in situ* measurements was initially evaluated globally. The upscaling coefficients displayed an acceptable overall accuracy, with 90% bias following a normal distribution within the range of ± 0.02. The performance of the upscaling model is significantly influenced by spatial heterogeneity. However, the impact of spatial diversity depends on the altitude and type of land coverage, and it becomes more significant as elevation increases and covers the land cover type of GRA.

It is important to note that the absolute truth on the coarse pixel scale is unattainable due to the limitations in instruments and measurement methods as well as the uncertainty of the upscaling model (Wu et al., 2019; Wen et al., 2022). Instead, the relative truth can be used to approximate the absolute truth. What can be done is to improve the accuracy of pixel scale relative truth (also denoted as "truth") as much as possible. For instance, the *in situ* measurements can be directly used as the pixel scale reference over homogeneous surfaces or in the case that the satellite acquisition and in situ measurement footprints are similar, and the upscaling model is not necessary as it has its own source of uncertainty. But the upscaling model is useful for heterogeneous areas when *in situ* measurement footprints are less than satellite pixel size, because it increases the representativeness of the sampling for direct validation. The accuracy assessment results of pixel scale ground "truth" dataset demonstrate that the accuracy of reference data can be enhanced by 17.09% over the regions with strong spatial heterogeneity. However, the degree of improvement with this dataset displays a decreasing trend as the reduction of spatial heterogeneity. At a global scale, the pixel scale ground "truth" dataset enhances the accuracy of pixel scale reference data in general, with the overall RRMSE decreased by 6.04% compared to *in situ* single site measurements.

Currently, a community-based validation tool, such as SALVAL (Sánchez-Zapero et al., 2023), could provide a framework for undertaking performance assessments through well-defined and uniform procedures, metrics and reference observations for all involved datasets, resulting in increased comparability, in addition to the ability to import new product datasets. Our dataset, obtained through standardized operational procedures, permits expanding established datasets to spatially underrepresented sites. This newly introduced dataset serves as a remedy to the inadequacy and inconsistency of the

reference data currently employed in validation/correction efforts, thereby paving the way for the coordinated use of various satellite albedo products and unlocking the full capacity of different albedo products.

**Author contributions.** FP, XW was responsible for the main research ideas and writing the manuscript. JW, DY, XL, and QX contributed to the data collection. XW, RT, QZ, and JW have supported the work with formal analysis, and XW contributed to the manuscript organization. All authors have worked on the writing, particularly for review and editing.

**Competing interests.** The contact author has declared that none of the authors has any competing interests.

**Disclaimer.** Publisher's note: Copernicus Publications remains neutral with regard to jurisdictional claims in published maps and institutional affiliations.

**Acknowledgements.** The authors would like to thank the various site networks and services, namely Fluxnet, BSRN, SURFRAD, HiwaterWSN and Huailai station, for their dedicated efforts in providing the ground measurements that have contributed to the generation of the coarse pixel scale "truth" dataset.

**Financial support.** This work was jointly supported by the National Natural Science Foundation of China (Grant No. 41971316 and 42071296) and the Fundamental Research Funds for the Central Universities (lzujbky-2020-72).

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
