# Peer review of "A coarse pixel scale ground "truth" dataset based on the global *in* situ site measurements to support validation and bias correction of satellite surface albedo products"

_Earth System Science Data, 2023_

## Referee Comment (RC1)

**GENERAL COMMENT**

The dataset based on upscaling could be very useful for the community, as it is a huge compilation of data from 368 sites mainly distributed in North hemisphere (mainly North of America and Europe). However, I miss a representativeness over Australia, where there are already available a large quantity of networks providing in situ albedo measurements.

Optimism about the fact of the need for upscaling techniques should be toned down, as it is not a need as community-agreed validation protocols recommend the use of in situ tower measurements, as they are the real 'truth'. The upscaling approach could be useful for heterogeneous areas, allowing increasing the representativeness of the sampling for direct validation at global scale. However, this approach introduces other sources of uncertainties, as the uncertainty of satellite high-resolution input is propagated and higher than in situ measurements. I miss this aspect in the dataset (the uncertainty should be provided). To be compliant with the concept of 'fiducial reference data', the uncertainties should be quantified and provided along with the reference dataset for conformity testing of satellite products. It is well-known that upscaling introduces additional sources of uncertainties. The next generation of satellites will reduce the spatial resolution of global coarse resolution products, allowing the use of point in situ data. Then, it should be discussed the originality of this datasets for future applications.

Based on the validation results of the method, the upscaling maps show similar uncertainty (RMSE) than existing albedo satellite products when they are compared with direct in situ measurements. Then, the upscaling method provides a useful approach to increase the number of sample for direct validation purpose but it cannot be considered as real 'truth'. This should be clearly demonstrated.

Additionally, I recommend reviewing the use of the English language along the manuscript. The presentation of the methods and results should be presented more clearly. It would be necessary to specify which datasets, quantities and resolutions (spatial and temporal) used in each step.

**SPECIFIC COMMENT**

**Title**

What do you mean by 'bias correction'? In situ measurements support validation of satellite products, providing useful data for bias quantification of satellite products. I am not sure how in situ measurement could be used to correct the bias of a satellite product.

**Absract**

Line 13: Same comment as before in regard to 'bias correction'. Justify the use of the 'correction term' or modify by bias quantification

Line 14: What satellite measurements are you referring? Low, medium or high (decametric) instruments.

Line 16: Justify the need for upscaling. If satellite acquisition and in situ measurement footprints are similar the upscaling introduces additional sources of uncertainties.

**Introduction**

Lines 55-56: The community-agreed surface albedo validation protocol (CEOS Working Group on Calibration and Validation – Land Product Validation subgroup) disagreed with this affirmation. Ground measurement can be directly used to validate satellite pixels. The current community-agreed approach is based on the evaluation of the spatial representativeness of ground measurement (Román et al., 2010, 2009).
Reference: CEOS LPV albedo protocol: https://lpvs.gsfc.nasa.gov/PDF/CEOS_ALBEDO_Protocol_20190307_v1.pdf

Lines 56-57: 'Limited by the means and methods of ground measurement, the absolute truth on the coarse pixel scale cannot be obtained.' Justify this sentence.

Lines 65-67: 'However, in situ measurements cannot be directly used as the coarse pixel scale truth given that the footprint of in situ sites is far less than the scale of a coarse pixel.' Please justify this or rephrase this sentence. In situ measurement footprint depends on the tower height. Depending of tower height and satellite spatial resolution they can be compared.

**Section 2.1**

Lines 115-118: 'These radiometers have been rigorously calibrated and continuously supervised to reduce systematic measurement errors (Jia et al., 2013; Wang et al., 2009; Zhou et al., 2016).' Are you confident that all radiometers from 368 sites have been rigorously calibrated and continuously supervised? This is not the case based on the references you are providing.

Lines 118-120: Justify the use of measurement at the local solar noon.

**Section 2.3**

The formula proposed to combine WSA and BSA used the diffuse light ratio, which is an approximation. The actual diffuse solar radiation should be used, as is the real model considering the actual environment (as you said), as considers the real atmospheric state.

Justify the use of this approximation, including the uncertainties introduced in this step. The limitations over snow targets should be also discussed.

I cannot find he formula used to calculate sky diffuse light ratio in the provided reference (Stokes and Schwartz (1994)). Please, use the right reference.

**Section 2.2**

Not clear what definition of satellite product is used according to illumination geometry (black-sky, white-sky)? Please provide more details about that.

**Section 2.4**

lines 177 – 189: This part does not correspond to ancillary data. Here you are describing the spatial heterogeneity metric (std) that should be moved to the 'methodology' section.

**Section 3.1**

 I miss a diagram clearly showing the process of the upscaling model.

**Section 4.1**

The performance of the upscaling model shows that the uncertainty (RMSE) of the upscaled maps is typically between 0.03 and 0.05, which is the typical uncertainty of the surface albedo coarse resolution satellite products (e.g., MCD43A3, GLASS, GlobAlbedo, C3S SPOT/VGT, C3S PROBA-V, C3S Sentinel-3). In conclusion, the uncertainty of the upscaled maps is similar to any other product and it is questionable its utility as a reference 'ground-truth'.

**Section 4.2**

It is not clear which albedo quantities are you comparing: albedo single site, albedo upscaling, reference? You should focus your discussion also based on the different albedo definitions of these quantities (blue-sky, black-sky, etc).

It is not clear the spatial coverage of the study. You should clearly indicate the spatial resolution related to all datasets used in this section: albedo single site, albedo upscaling, reference.

**Section 4.3**

The validation of MCD43A3 V0061 using pixel scale ground 'truth' is only presented for some sites. The selection of these sites (and not others) should be justified.

What is the reason of large differences (outliers) over CA-NS2, CA-LP1, IT-Tor ?

Additionally, I miss the overall figure using the whole dataset.

**Conclusions**

There already exist other initiatives, like GBOV (https://gbov.acri.fr/), providing similar datasets to that presented in this manuscript, and should be mentioned.

On the other case, during the manuscript there are comments related to lack of standardized methods and operational validation systems for albedo validation. In fact, the CEOS/WGCV LPV subgroup (https://lpvs.gsfc.nasa.gov/) is coordinating these activities. An operational validation system was recently endorsed by CEOS/WGCV LPV, which is called SALVAL (Sánchez-Zapero et al., 2023) and it allows albedo products to reach operational and globally representative validation results (CEOS LPV stage 4). Access to SALVAL is available on https://calvalportal.ceos.org/salval

**TECHNICAL CORRECTIONS**

**Line 18 – Abstract**

'in situ' is not hyphenated. Please review the whole manuscript to homogenize 'in situ' term.

**Line 64 – Introduction**

Remove '.' before references

**Line 145**

'ith'?

**Bibliography**

Román, M.O., Schaaf, C.B., Lewis, P., Gao, F., Anderson, G.P., Privette, J.L., Strahler, A.H., Woodcock, C.E., Barnsley, M., 2010. Assessing the coupling between surface albedo derived from MODIS and the fraction of diffuse skylight over spatially-characterized landscapes. Remote Sens. Environ. 114, 738–760. https://doi.org/10.1016/j.rse.2009.11.014

Román, M.O., Schaaf, C.B., Woodcock, C.E., Strahler, A.H., Yang, X., Braswell, R.H., Curtis, P.S., Davis, K.J., Dragoni, D., Goulden, M.L., Gu, L., Hollinger, D.Y., Kolb, T.E., Meyers, T.P., Munger, J.W., Privette, J.L., Richardson, A.D., Wilson, T.B., Wofsy, S.C., 2009. Remote Sensing of Environment The MODIS ( Collection V005 )

BRDF / albedo product : Assessment of spatial representativeness over forested landscapes. Remote Sens. Environ. 113, 2476–2498. https://doi.org/10.1016/j.rse.2009.07.009

Sánchez-Zapero, J., Martínez-Sánchez, E., Camacho, F., Wang, Z., Carrer, D., Schaaf, C., García-Haro, F.J., Nickeson, J., Cosh, M., 2023. Surface ALbedo VALidation (SALVAL) Platform: Towards CEOS LPV Validation Stage — Application to Three Global Albedo Climate Data Records. Remote Sens. 2023, Vol. 15, Page 1081 15, 1081. https://doi.org/10.3390/RS15041081

---

## Referee Comment (RC2)

Review of "A coarse pixel scale ground "truth" dataset based on the global in situ site measurements to support validation and bias correction of satellite surface albedo products" by Fei Pan et al.

The authors constructed a global albedo database in coarse pixel scale based on the high-resolution Landsat7 ETM+ images and 368 in situ sites from sparsely distributed observation networks globally. The results showed that the new database overcomes the shortcoming of in situ albedo measurements and can be used as ground truth, which captures spatiotemporal variations of surface albedo. However, there are many mistakes in the current manuscript which are due to the carelessness of the authors. Moreover, some parts of the content have indications of plagiarism. Therefore, before the current manuscript can be published, the authors should reply to the following comments diligently.

**Major comments:**
1. As described by the authors, one criterion of the methodology in this manuscript is the spatial resolution of high-resolution albedo observation should be equivalent to the footprint of in-situ observation (lines 205-207). However, the authors also highlighted that the footprints of in-situ sites are not fixed. It depends on the height of the albedometers (Lines 113-115). Have the authors compared the size of the footprints of a total of 368 in-situ sites with that of the Landsat7 ETM+ (30 m)? How about the results? Please discuss this issue with figures or tables.
2. In the manuscript, the coarse spatial resolution of the albedo product is 500 m (MCD43A3 V061) and the high resolution of the albedo is 30 m (Landsat7 ETM+). Therefore, the authors retrieved the upscaling coefficients to upscale the surface albedo from a high resolution of 30 m to a coarse resolution of 500 m. However, since 500 cannot be divided by 30, there should be some high-resolution observations partially covered at the edge of the coarse grid. How to deal with this issue? Please explain.
3. Figure 3: The label of x-axis is wrong. According to line 292, Fig. 3 is the scatter plot of $\theta_{upscaling}$ and $\theta_{reference}$, none of them should be the "Pixel scale ground truth". Please check.
   Meanwhile, the six subpanels represented six land cover types according to the caption of Fig. 3. However, the authors didn't mention their locations (Lon/Lat) as well as the land cover types. Please add.
4. Figure 4: Please add line x=0 in the subpanel of Bias. Meanwhile, I don't agree with the expression "the biases concentrated around 0" in the conclusion (Line 482). Please revise the relevant content.

5. Figures 5-9: I cannot find the description of the mean of the boxplot. What is the meaning of the line in the center box? The mean of median value? Please describe it clearly.
   Meanwhile, what's the sample number of each boxplot? Please add the description and tables.

6. Lines 225-226: how to choose the $\theta_{ETM+\_in\_situ}$? Do you mean the nearest Landsat7 ETM+ pixel to the in situ site? Please explain.

7. According to Fig. 1, there is a large portion of regions without in situ sites, especially for the regions covered with snow (e.g., Siberia) or with high elevation (e.g., Tibet). Therefore, how can the authors announce that their database can be used globally (in the abstract and conclusion)? Please explain.

8. Results and Discussion: The bias and RMSE of the upscaling results seems equivalent to the typical uncertainty of the surface albedo coarse resolution satellite products. Why are the authors satisfied with their results? Please explain.

9. Methodology: The content and the structure of the methodology in the current manuscript are quite similar to those of Wu et al., (2020). I also find the reference "Peng et al. (2015)" in line 240 is not included in the References part of the current manuscript. So, I believe the author who wrote the current manuscript plagiarized the whole content of methodology from Wu et al., (2020) and just modified some keywords. I leave the decision to the editor to decide whether to reject the current manuscript.

10. The current manuscript should be polished before resubmission.

**Minor comments:**
1. Please check the number of equations throughout the manuscript. I found **two** "equation (4)" and "equations (10-12)". Moreover, I found the size of the equation numbers is different. Please explain the reason.

2. Line 220: the right side of this equation is wrong. A comma is missing in the subscript. Please refer to the paper Wu et al., (2020), and fix it.

3. Line 226: the size of the words "indicates the" is smaller than the others, please explain the reason.

4. Line 237: What does the $\theta_{in\ situ}$ stand for? Please describe it in the main content clearly.

5. Line 240: I cannot find the reference "Peng et al. (2015)" in your "References".

6. Line 273: the metric "coefficient of determination ($R^2$)" was introduced in line 269, but the equation only gave "R". Please explain the reason.

7.  Line 292: Please make sure it is "Fig.2" or "Fig. 3"? The same problem also can be found in line 299 (Fig. 3 or Fig. 4).
8.  Line 335: the lowest RMSE around "0.3"? Are you sure?

**References:**

X. Wu, J. Wen, Q. Xiao and D. You, "Upscaling of Single-Site-Based Measurements for Validation of Long-Term Coarse-Pixel Albedo Products," in *IEEE Transactions on Geoscience and Remote Sensing*, vol. 58, no. 5, pp. 3411-3425, May 2020, doi: 10.1109/TGRS.2019.2954879.

---

## Author Comment (AC1)

**Response to comments**

**Paper #:** essd-2023-220

**Title:** A coarse pixel scale ground "truth" dataset based on the global in situ site measurements to support validation and bias correction of satellite surface albedo products

5  **Journal:** Earth System Science Data

Thank you for providing us with so many valuable suggestions and they do help improve the paper. According to the reviewers' comments and suggestions, we revised the paper carefully and tried to give satisfactory answers to the reviewers' questions. The corresponding modifications are highlighted in red

10  font in the revised paper.

The summaries of the revision for this paper are as follows:

First, we have reorganized the data and added all available sites. Moreover, parts of results and discussion, main findings and conclusion, as well as the abstract were rewritten based on the complete dataset.

15  Second, the necessity for upscaling models was further elucidated by integrating the work of other researchers in **Introduction** and **Conclusion**. Furthermore, we discussed the applicability of upscaling models at various sites and provided an objective statement about the role and significance of the pixel scale ground "truth" dataset. Its relationship with existing satellite albedo products and ground measurements was also explained.

20  Third, we have added the quantification of uncertainty of upscaling models for each site in **Section 4.1**. Moreover, we have described how we addressed the issue of varying footprint sizes at distinct sites, as well as the rationale for implementing ETM+ imagery.

Fourth, we have explained the spatial and temporal resolution of the different data used in the methodology and conclusions, and added a detailed description of the illumination geometry, including

25  black-sky albedo, and white-sky albedo, for the albedo products used. Additionally, we have clarified the sample size for the boxplots and re-examined the implications regarding sample size in the **Results** and **Discussion** section.

Fifth, we have explained the reason for the methodology Section being similar to those of Wu et al.(2020), and emphasized the importance of the content of our work.

30  Sixth, we have corrected typing errors; complemented supporting evidence and literature; improved charts and figures; and corrected spelling and grammatical errors in this paper.

For the specific comments for each reviewer, we have made a detailed reply as follows.

**Reviewer #2**

35  The authors constructed a global albedo database in coarse pixel scale based on the high-resolution Landsat7 ETM+ images and 368 *in situ* sites from sparsely distributed observation networks globally. The results showed that the new database overcomes the shortcoming of *in situ* albedo measurements and can be

used as ground truth, which captures spatiotemporal variations of surface albedo. However, there are many mistakes in the current manuscript which are due to the carelessness of the authors. Moreover, some parts of the content have indications of plagiarism.

Therefore, before the current manuscript can be published, the authors should reply to the following comments diligently.

As described by the authors, one criterion of the methodology in this manuscript is the spatial resolution of high-resolution albedo observation should be equivalent to the footprint of in-situ observation (lines 205-207). However, the authors also highlighted that the footprints of in-situ sites are not fixed. It depends on the height of the albedometers (Lines 113-115). Have the authors compared the size of the footprints of a total of 368 in-situ sites with that of the Landsat7 ETM+ (30 m)? How about the results? Please discuss this issue with figures or tables.

Re: The footprint of *in situ* sites is a function of measurement heights of the albedometers from the underlying surface and the field of view of the sensors. The former typically depends on the height of tower and height of the canopy top (different at different time), which are generally different from one site to another. The latter is not fully consistent due to the ideal and non-ideal cosine response of the sensors (Balzarolo et al., 2011; Cescatti et al., 2012; Song et al., 2019; Marion, 2021). Therefore, the footprints of *in situ* sites are not fixed. However, it is difficult to make a comparison between the footprints of *in situ* observation and the spatial resolution of high-resolution albedo observation. Because the footprints of *in situ* sites are various. Even for the same site, the footprint of *in situ* site is not consistent at different time due to the change of underlying surface (e,g., vegetation growth). But the effect of the spatial scale difference between *in situ* measurements and high-resolution data is believed to be negligible since the selection of high resolution data follows strict rules:

First, its spatial resolution should be minimal to maintain surface homogeneity within the fine pixel scale and ensure stable radiation acquisition.

Second, according to the albedo data observed at the FLUXNET site, approximately 80% of the energy in the observed signal originates from within 10-20 meters of the flux tower (Cescatti et al. 2012; Wang et al., 2014). Hence, the spatial resolution of the data should be near the footprint of *in situ* sites.

Third, since the upscaling coefficients were determined by long-time series high-resolution albedo maps and then were applied to long time series *in situ* measurements, the high-resolution albedo maps should cover at least one full cycle period, typically a year, to account for seasonal changes in surface heterogeneity caused by phenology and to guarantee the stability of the upscaling coefficients.

For these reasons, the Landsat ETM+ albedo data were adopted in this study. In the revised manuscript, we have added these explanations in **Section 3.1**.

In the manuscript, the coarse spatial resolution of the albedo product is 500 m (MCD43A3 V061) and the high resolution of the albedo is 30 m (Landsat7 ETM+). Therefore, the authors retrieved the upscaling coefficients to upscale the surface albedo from a high resolution of 30 m to a coarse resolution of 500 m. However, since 500 cannot be divided by 30, there should be some high-resolution observations partially covered at the edge of the coarse grid. How to deal with this issue? Please explain.

Re: In fact, we have used the 17×17 ETM+ pixels (an approximately 510 m ×510 m area) centered at MODIS pixel to calculate the pixel scale ground "truth". Namely, the spatial resolution of the ground "truth" is 510 m. The difference between the spale scale of MCD43A3 V061 and pixel scale ground "truth" is negligibly small, because the spatial response is very small at the margin areas of the pixel (Peng et al., 2015). To clarify this point, we have added the sentence as "*Secondly, it facilitated coarse pixel-level aggregation within a 17×17 window (an approximately 510 m ×510 m area, considered as a coarse scale pixel), serving to be the reference value of the coarse pixel albedo.*" in **Section 2.2**.

[Figure]

**Figure. The point spread function of MODIS albedo products (Peng et al., 2015).**

**References:**

Peng, J., Liu, Q., Wang, L., Liu, Q., Fan, W., Lu, M., and Wen, J.: Characterizing the Pixel Footprint of Satellite Albedo Products Derived from MODIS Reflectance in the Heihe River Basin, China, Remote Sensing, 7(6), 6886-6907, https://doi.org/10.3390/rs70606886, 2015.

Figure 3: The label of x-axis is wrong. According to line 292, Fig. 3 is the scatter plot of $\theta_{upscaling}$ and $\theta_{reference}$, none of them should be the "Pixel scale ground truth". Please check.

Re: Great thanks for pointing out this mistake. The mistake has been corrected as:

[Figure]

**Figure 3: The scatter plots between the upscaling results ($\theta_{upscaling}$ ) with the upscaling models and the coarse pixel scale reference ($\theta_{reference}$ ).**

Meanwhile, the six subpanels represented six land cover types according to the caption of Fig. 3. However, the authors didn't mention their locations (Lon/Lat) as well as the land cover types. Please add.

Re: We have added information about the *in situ* sites that correspond to the six subpanels in **Section 4.1.**

**Table 1: Description of the *in situ* sites used in the model performance analysis.**

| Networks | US-UMB | CA-NS2 | US-Ha2 | FR-Gri | CA-Lp1 | IT-Tor |
|---|---|---|---|---|---|---|
| Location(lon, lat) | (-84.7138, 45.5598) | (-98.5247, 55.9058) | (-72.1779, 42.5393) | (13.51259, 50.9500) | (-122.8414, 55.1119) | (7.5781, 45.8444) |
| Spatial heterogeneity | 0.0133079 | 0.0640852 | 0.0065224 | 0.5564959 | 0.18694994 | 1.01929451 |
| Elevation(m) | 236.72682 | 271.09771 | 367.29669 | 377.65914 | 749.265564 | 2162.78979 |
| Land cover type | DBF | EBF | MF | CRO | WSA | GRA |

Figure 4: Please add line x=0 in the subpanel of Bias. Meanwhile, I don't agree with the expression "the biases concentrated around 0" in the conclusion (Line 482). Please revise the relevant content.

Re: As suggested by the reviewer, the line x=0 in the subpanel of Bias in Figure 4 has been added.

[Figure]

**Figure 4. Distribution of RMSE (a), Bias (b), and R² (c) of the upscaling coefficients. The histograms presented here combine the results of the 416 *in situ* sites.**

The expression "*the biases concentrated around 0*" in the Conclusion has been revised. The related sentence has been rephrased as "*The suitability of the upscaling model for applying to the in situ measurements was initially evaluated globally. The upscaling coefficients displayed an acceptable overall accuracy, with 90 % of bias following a normal distribution within the range of ± 0.02.* ".

Figures 5-9: I cannot find the description of the mean of the boxplot. What is the meaning of the line in the center box? The mean of median value? Please describe it clearly.

Re: As suggested by the reviewer, we have added the description of the mean of the boxplot. The black lines denote the median values. Taking Figure 6 as an example, the revised figure is shown as follows.

[Figure]

Figure 6: Boxplots showing the dependence of RMSE (a) and $R^2$(b) of the upscaled albedo on spatial heterogeneity. Three different degrees of spatial heterogeneity are marked by different colors. Black lines indicate median values. Outliers are values that are farther than 1.5 interquartile ranges. The accuracy response of the upscaling model to different spatial heterogeneity. The number of *in situ* sites with spatial heterogeneity of [0,0.1], [0.1-0.3], and [0.3-1.5] are 337, 49, and 30, respectively.

Meanwhile, what's the sample number of each boxplot? Please add the description and tables.

Re: In the revised manuscript, we have added the number of *in situ* sites for each level of spatial heterogeneity (Figure 6), each level of elevation (Figure 7), and each land cover type(Figure 9).

Lines 225-226: how to choose the $\theta_{ETM+\_in\_situ}$? Do you mean the nearest Landsat7 ETM+ pixel to the *in situ* site? Please explain.

Re: $\theta_{ETM+\_in\ situ}$ denotes the ETM+ pixel albedo time series containing the *in situ* site. Namely, it refers to the ETM+ pixel in which *in situ* site is located.

According to Fig. 1, there is a large portion of regions without *in situ* sites, especially for the regions covered with snow (e.g., Siberia) or with high elevation (e.g., Tibet). Therefore, how can the authors announce that their database can be used globally (in the abstract and conclusion)? Please explain.

Re: In the revised manuscript, we have added the *in situ* albedo measurements over Australia in the revised manuscript. Moreover, the *in situ* measurements over Siberia and other regions with effective measurements were also included in the dataset. The number of *in situ* sites increased to 416 for the dataset. It is true that the number of *in situ* sites is more than 416 within the globe. However, some sites were

145 excluded either due to the lack of incoming radiation information or the small data size after quality control. The distribution of these *in situ* sites is shown as follows. Given that these *in situ* sites are widely distributed on the globe and cover a wide range of environmental conditions (atmospheric model, aerosol model, spatial homogeneity and heterogeneity, temporal variation characteristics), they were believed to be representative of the globe.

[Figure]

150

**Figure 1. The distribution of the 416 *in situ* sites over different land cover types.**

Results and Discussion: The bias and RMSE of the upscaling results seems equivalent to the typical uncertainty of the surface albedo coarse resolution satellite products. Why are the authors satisfied with
155 their results? Please explain.

Re: It is true that the upscaling model itself has errors because it suffers from its own source of uncertainty. Therefore, over homogeneous surfaces where *in situ* site measurements are spatially representative, using this upscaling model will bring no benefits or even counteract due to the errors of the upscaling model. Nevertheless, over heterogeneous surface where *in situ* sites are lack of spatial
160 representativeness, the benefits outweigh disadvantages. The accuracy assessment results of pixel scale ground "truth" dataset demonstrate that the accuracy of reference data can be enhanced by 17.09 % over the regions with strong spatial heterogeneity. However, the degree of improvement with this dataset displays a decreasing trend as the reduction of spatial heterogeneity. In order to clarify this point, we have added the paragraph "*……For instance, the in situ measurements can be directly used as the pixel scale reference over*
165 *homogeneous surfaces or in the case that the satellite acquisition and in situ measurement footprints are similar, and the upscaling model is not necessary as it has its own source of uncertainty. But the upscaling model is useful for heterogeneous areas when in situ measurement footprints are less than satellite pixel size, because it increases the representativeness of the sampling for direct validation. The accuracy assessment results of pixel scale ground "truth" dataset demonstrate that the accuracy of reference data can be*

*enhanced by 17.09 % over the regions with strong spatial heterogeneity……"* in **Conclusion**.

As regards to the accuracy of the current coarse resolution surface albedo satellite products, their accuracy (between 0.03 and 0.05) is usually assessed over relatively homogeneous land surfaces. And the validation works over heterogeneous are still rare currently. The spatial scale mismatch over heterogeneous surfaces remains to be challenging to fully understand the overall accuracy of satellite products in different areas. Hence, our dataset can be considered as an important addition to the reference data on the coarse pixel scale over heterogeneous land surfaces.

Methodology: The content and the structure of the methodology in the current manuscript are quite similar to those of Wu et al., (2020). I also find the reference "Peng et al. (2015)" in line 240 is not included in the References part of the current manuscript. So, I believe the author who wrote the current manuscript plagiarized the whole content of methodology from Wu et al., (2020) and just modified some keywords. I leave the decision to the editor to decide whether to reject the current manuscript.

Re: We really appreciate the rigorous scientific attitude of the reviewer. In fact, the upscaling methodology of Wu et al., (2020) was developed by our research group, and the authors of Wu et al. (2012) are also the main contributors to this paper. However, the paper of Wu et al. (2020) merely proposed the upscaling method and did not comprehensively assess the effectiveness of this upscaling method. Moreover, this upscaling method has never been applied to the single *in situ* site measurements of the sparsely globally distributed observation networks (e.g., SURFRAD, BSRN, and Fluxnet) except for Huailai and Heihe River Basin, China. As a result, its transferability to *in situ* sites all over the world is still unknown. As the continuation and deepening of our previous work (Wu et al., 2020), this study puts emphasis on the comprehensive evaluation and extensive use of this upscaling method. Furthermore, a coarse pixel scale ground "truth" dataset was provided for validation and bias correction of satellite surface albedo products.

To counter and prevent misunderstanding, we have added the sentence as *"To overcome the representative errors of in situ measurements and promote utilization ratio of in situ sites from these sparse networks in validation, Wu et al. (2020) have proposed an upscaling method specified for the single site in situ measurements. However, the effectiveness of this method has not been comprehensively assessed and its transferability to in situ sites all over the world is still unknown. As the continuation and deepening of our previous work (Wu et al., 2020), this study puts emphasis on the comprehensive evaluation and extensive use of this upscaling method based on 416 in situ sites throughout the world. Furthermore, a coarse pixel scale ground "truth" dataset was provided for validation and bias correction of satellite surface albedo products. The potential usage of this dataset was also discussed."* in **Introduction** of the revised manuscript.

The reference of Peng et al. (2015) has been added to the reference list.

The current manuscript should be polished before resubmission.

Re: Great thanks for the comment. The manuscript has been polished by a native speaker.

**Minor comments:**

Please check the number of equations throughout the manuscript. I found two "equation (4)" and "equations (10-12)". Moreover, I found the size of the equation numbers is different. Please explain the reason.

Re: We have corrected these errors in the revised manuscript.

Line 220: the right side of this equation is wrong. A comma is missing in the subscript. Please refer to the paper Wu et al., (2020), and fix it.

Re: This mistake has been corrected.

Line 226: the size of the words "indicates the" is smaller than the others, please explain the reason.

Re: The font size has been made consistent.

Line 237: What does the $\theta_{in\ situ}$ stand for? Please describe it in the main content clearly.

Re: $\theta_{in\ situ}$ denotes *in situ* site measurement. To describe it more clearly, this sentence has been revised as "*When the upscaling coefficients were determined, they were applied to in situ site measurements ($\theta_{in\ situ}$) to simulate the in situ reporting of surface albedo ($\theta_{in\ situ\_ETM+}$)……*" in the revised manuscript.

Line 240: I cannot find the reference "Peng et al. (2015)" in your "References".

Re: The reference of Peng et al. (2015) has been added to the reference list.

Line 273: the metric "coefficient of determination (R2)" was introduced in line 269, but the equation only gave "R". Please explain the reason.

Re: The coefficient of determination ($R^2$) was employed in this paper. The equation (15) has been revised as:

$$R^2 = \frac{[\sum_{d=1}^{L}(\theta_{upscaling}(d) - \overline{\theta_{upscaling}})(\theta_{reference}(d) - \overline{\theta_{reference}})]^2}{\sum_{d=1}^{L}(\theta_{upscaling}(d) - \overline{\theta_{upscaling}})^2 \sum_{d=1}^{l}(\theta_{reference}(d) - \overline{\theta_{reference}})^2} \tag{15}$$

Line 292: Please make sure it is "Fig.2" or "Fig. 3"? The same problem also can be found in line 299 (Fig. 3 or Fig. 4).

Re: The formulation (e.g., Fig. 2, Fig. 3, Fig. 4) has been made consistent throughout the paper.

Line 335: the lowest RMSE around "0.3"? Are you sure?

Re: We are sorry for this mistake. It should be 0.03. We have thoroughly checked the revised manuscript to avoid typos.

---

## Author Comment (AC2)

**Response to comments**

**Paper #:** essd-2023-220

**Title:** A coarse pixel scale ground "truth" dataset based on the global in situ site measurements to support validation and bias correction of satellite surface albedo products

5 **Journal:** Earth System Science Data

Thank you for providing us with so many valuable suggestions and they do help improve the paper. According to the reviewers' comments and suggestions, we revised the paper carefully and tried to give satisfactory answers to the reviewers' questions. The corresponding modifications are highlighted in red

10 font in the revised paper.

The summaries of the revision for this paper are as follows:

First, we have reorganized the data and added all available sites. Moreover, parts of results and discussion, main findings and conclusion, as well as the abstract were rewritten based on the complete dataset.

15 Second, the necessity for upscaling models was further elucidated by integrating the work of other researchers in **Introduction** and **Conclusion**. Furthermore, we discussed the applicability of upscaling models at various sites and provided an objective statement about the role and significance of the pixel scale ground "truth" dataset. Its relationship with existing satellite albedo products and ground measurements was also explained.

20 Third, we have added the quantification of uncertainty of upscaling models for each site in **Section 4.1**. Moreover, we have described how we addressed the issue of varying footprint sizes at distinct sites, as well as the rationale for implementing ETM+ imagery.

Fourth, we have explained the spatial and temporal resolution of the different data used in the methodology and conclusions, and added a detailed description of the illumination geometry, including

25 black-sky albedo, and white-sky albedo, for the albedo products used. Additionally, we have clarified the sample size for the boxplots and re-examined the implications regarding sample size in the **Results** and **Discussion** section.

Fifth, we have explained the reason for the methodology Section being similar to those of Wu et al.(2020), and emphasized the importance of the content of our work.

30 Sixth, we have corrected typing errors; complemented supporting evidence and literature; improved charts and figures; and corrected spelling and grammatical errors in this paper.

For the specific comments for each reviewer, we have made a detailed reply as follows.

**Reviewer #1**

35 The dataset based on upscaling could be very useful for the community, as it is a huge compilation of data from 368 sites mainly distributed in North hemisphere (mainly North of America and Europe). However, I miss a representativeness over Australia, where there are already available a large quantity of networks

providing *in situ* albedo measurements.

Re: Great thanks for the positive comments. We have added the *in situ* albedo measurements over Australia in the revised manuscript. Moreover, the *in situ* measurements over Siberia and other regions with effective measurements were also included in the dataset. The number of *in situ* sites increased to 416 for the dataset. The distribution of these *in situ* sites is shown as follows:

[Figure]

**Figure 1: The distribution of the 416 *in situ* sites over different land cover types.**

Optimism about the fact of the need for upscaling techniques should be toned down, as it is not a need as community-agreed validation protocols recommend the use of *in situ* tower measurements, as they are the real 'truth'. The upscaling approach could be useful for heterogeneous areas, allowing increasing the representativeness of the sampling for direct validation at global scale.

Re: We are sorry for not making it clear to readers. As pointed out by the reviewer, the upscaling approach is useful for heterogeneous areas as it increases the representativeness of the sampling for direct validation. But it is not necessary over homogeneous land surfaces because *in situ* measurements are spatially representative in this case, and the utilization of upscaling model does not bring benefits as the upscaling model itself has uncertainty.

In order to clarify this point, we have added a paragraph as "*It is important to note that the absolute truth on the coarse pixel scale is unattainable due to the limitations in instruments and measurement methods as well as the uncertainty of the upscaling model (Wu et al., 2019; Wen et al., 2022). Instead, the relative truth can be used to approximate the absolute truth. What can be done is to improve the accuracy of pixel scale relative truth (also denoted as "truth") as much as possible. For instance, the in situ measurements can be directly used as the pixel scale reference over homogeneous surfaces or in the case*

*that the satellite acquisition and in situ measurement footprints are similar, and the upscaling model is not necessary as it has its own source of uncertainty. But the upscaling model is useful for heterogeneous areas when in situ measurement footprints are less than satellite pixel size, because it increases the representativeness of the sampling for direct validation. The accuracy assessment results of pixel scale ground "truth" dataset demonstrate that the accuracy of reference data can be enhanced by 17.09 % over the regions with strong spatial heterogeneity. However, the degree of improvement with this dataset displays a decreasing trend as the reduction of spatial heterogeneity. At a global scale, the pixel scale ground "truth" dataset enhances the accuracy of pixel scale reference data in general, with the overall RRMSE decreased by 6.04 % compared to in situ single site measurements.*" in **Conclusion**.

However, this approach introduces other sources of uncertainties, as the uncertainty of satellite high-resolution input is propagated and higher than *in situ* measurements. I miss this aspect in the dataset (the uncertainty should be provided).

Re: It is true that the upscaling model has its own source of uncertainty. As recommended by the reviewer, we have added the information on the uncertainty of the upscaling approach for each site in **Section 4.1** as follows. The specific values of the uncertainty of the upscaling model have been shown in the file at the link to the dataset, where each site is quantified separately.

[Figure]

Figure 5: Spatial distribution of RMSE (a) and $R^2$ (b) of the upscaling model.

80  To be compliant with the concept of 'fiducial reference data', the uncertainties should be quantified and provided along with the reference dataset for conformity testing of satellite products.

Re: Thanks very much for this good suggestion. The uncertainty of the dataset has been quantified and provided along with the reference dataset as we explained above.

85  It is well-known that upscaling introduces additional sources of uncertainties. The next generation of satellites will reduce the spatial resolution of global coarse resolution products, allowing the use of point *in situ* data. Then, it should be discussed the originality of this datasets for future applications.

Re: It is true that the next generation of satellites will allow the generation of high-resolution products which are comparable to *in situ* data. But the current coarse resolution products record the information in 90  the past and will serve as an important component to form the long time series of satellite data, which is quite important to study global change from a long-term perspective. Hence, this dataset is still useful to

validate or correct the errors of these coarse resolution satellite albedo products.

Based on the validation results of the method, the upscaling maps show similar uncertainty (RMSE) than existing albedo satellite products when they are compared with direct *in situ* measurements. Then, the upscaling method provides a useful approach to increase the number of sample for direct validation purpose but it cannot be considered as real 'truth'. This should be clearly demonstrated.

Re: Yes, our dataset is relative truth, not absolute truth. In fact, the absolute truth on the coarse pixel scale is unattainable due to the limitations in instruments and measurement methods as well as the uncertainty of the upscaling model (Wu et al., 2019; Wen et al., 2022). Instead, the relative truth can be used to approximate the absolute truth. This point has been clearly demonstrated in the revised manuscript as "*It is important to note that the absolute truth on the coarse pixel scale is unattainable due to the limitations in instruments and measurement methods as well as the uncertainty of the upscaling model (Wu et al., 2019; Wen et al., 2022). Instead, the relative truth can be used to approximate the absolute truth. What can be done is to improve the accuracy of pixel scale relative truth (also denoted as "truth") as much as possible. For instance, the in situ measurements can be directly used as the pixel scale reference over homogeneous surfaces or in the case that the satellite acquisition and in situ measurement footprints are similar, and the upscaling model is not necessary as it has its own source of uncertainty. But the upscaling model is useful for heterogeneous areas when in situ measurement footprints are less than satellite pixel size, because it increases the representativeness of the sampling for direct validation.*" in **Conclusion**.

Additionally, I recommend reviewing the use of the English language along the manuscript. The presentation of the methods and results should be presented more clearly. It would be necessary to specify which datasets, quantities and resolutions (spatial and temporal) used in each step.

Re: Thanks for your nice suggestion. The language of the paper has been polished by a native speaker. Regarding the specific information about the dataset used in each step, we have summarized this information as tables.

**Table 1**. The information on the data used in the upscaling process

| Symbols | Meaning | Spatial resolution | Temporal resolution |
| --- | --- | --- | --- |
| $\theta_{ETM+\_in\ situ}$ | ETM+ pixel albedo time series corresponding to *in situ* site | 30 m | Daily data throughout the whole time series (i.e., 2012-2018). |
| $\theta_{ETM+}$ | ETM+ pixel albedo at other areas within a coarse pixel | 30 m | Daily data throughout the whole time series (i.e., 2012-2018). |
| $\theta_{in\ situ\_ETM+}$ | In situ reporting of surface albedo for each ETM+ pixel within a coarse pixel | 30 m | Daily data throughout the whole time series (i.e., 2000-2021). |

| | | | |
|---|---|---|---|
| $\theta_{\text{in situ}}$ | In situ albedo measurement | with varying spatial resolution but near the ETM+ pixel scale | Daily data throughout the whole time series (i.e., 2000-2021). |

**Table 2.** The information on the data used in the evaluation of the upscaling model

| Symbols | Meaning | Spatial resolution | Temporal resolution |
|---|---|---|---|
| $\theta_{\text{RETM+}}$ | ETM+ simulated pixel albedo based on upscaling coefficients and $\theta_{ETM+}$ | 30 m | Daily data throughout the whole time series (i.e., 2019-2021). |
| $\theta_{\text{ETM+}}$ | the ETM+ pixel albedo containing *in situ* site | 30 m | Daily data throughout the whole time series (i.e., 2019-2021). |
| $\theta_{\text{upscaling}}$ | upscaling results based on the $\theta_{ETM+}$ and upscaling coefficients | 500 m | Daily data throughout the whole time series (i.e., 2019-2021). |
| $\theta_{\text{reference}}$ | reference coarse pixel scale albedo | 500 m | Daily data throughout the whole time series (i.e., 2019-2021). |

120

**Table 3.** The information of the data used in the assessment of coarse pixel scale ground "truth"

| Symbols | Meaning | Spatial resolution | Temporal resolution |
|---|---|---|---|
| $\theta_{\text{in situ\_ref}}$ | coarse pixel scale ground "truth" dataset | 500 m | Daily data throughout the whole time series (i.e., 2000-2021). |
| $\theta_{\text{reference}}$ | reference coarse pixel scale albedo | 500 m | Daily data throughout the whole time series (i.e., 2000-2021). |

**Specific comments**

What do you mean by 'bias correction'? *In situ* measurements support validation of satellite products,

125 providing useful data for bias quantification of satellite products. I am not sure how *in situ* measurement could be used to correct the bias of a satellite product.

Re: "Bias correction" is a statistical technique used in data analysis. It is employed to rectify systematic errors, commonly known as biases, in a dataset. These errors can stem from several sources, such as sensor inaccuracies, measurement methods, or modeling assumptions. The objective of bias

130 correction is to enhance data precision and reliability by eliminating or minimizing these systematic errors.

Since the pixel scale ground "truth" dataset has been established, on one hand, it can be used to assess the errors of satellite products; on the other hand, it can correct these errors through the models such as random forests, cumulative distribution function, and Kalman filter. For further reading on bias correction, the related articles can be seen below:

**References**:

Atiah, W. A., Johnson, R., Muthoni, F. K., Tsidu, G. M., Amekudzi, . K., Kwabena, O., and Kizito, F.: **Bias Correction** and Spatial Disagregation of Satellite-Based Data for the Detection of Rainfall Seasonality Indices, Heliyon, 9, e17604, http://dx.doi.org/10.2139/ssrn.4349361, 2023.

Wang, J., Wu, X., Tang, R., Zeng, Q., Li, Z., Wen, J., and Xiao, Q.: Evaluation of three **error-correction** models based on the matched pixel scale ground "truth": A case study of MCD43A3 V006 over the Heihe River Basin, China, IEEE Journal of Selected Topics in Applied Earth Observations and Remote Sensing, 15, 8785-8797, https://doi.org/10.1109/JSTARS.2022.3213184, 2022.

Iqbal, Z., Shahid, S., Ahmed, K., Wang, X., Ismail, T., and GGabriel, H. F.: **Bias correction** method of high-resolution satellite-based precipitation product for Peninsular Malaysia, Theoretical and Applied Climatology, 148, 1429–1446, https://doi.org/10.1007/s00704-022-04007-6, 2022.

Katiraie-Boroujerdy, P.-S., Rahnamay Naeini, M., Akbari Asanjan, A., Chavoshian, A., Hsu, K.-L., and Sorooshian, S.: **Bias Correction** of Satellite-Based Precipitation Estimations Using Quantile Mapping Approach in Different Climate Regions of Iran, Remote Sensing, 12, 2102, https://doi.org/10.3390/rs12132102, 2020.

Line 13: Same comment as before in regard to 'bias correction'. Justify the use of the 'correction term' or modify by bias quantification.

Re: Explained in the previous question.

Line 14: What satellite measurements are you referring? Low, medium or high (decametric) instruments.

Re: It refers to satellite data with low spatial resolution.

Line 16: Justify the need for upscaling. If satellite acquisition and *in situ* measurement footprints are similar the upscaling introduces additional sources of uncertainties.

Re: It is true that the upscaling introduces additional sources of uncertainties if satellite acquisition and *in situ* measurement footprints are similar. In the revised manuscript, this sentence has been revised as "*The results demonstrate that using this dataset in validation outperforms the direct comparison between satellite and in situ site measurements over heterogeneous surfaces when in situ measurement footprints are less than satellite pixel size.*".

Furthermore, we have made it clear that the upscaling model is not necessary over homogeneous surfaces or in the case that the satellite acquisition and *in situ* measurement footprints are similar in **Conclusion** as "…....*the in situ measurements can be directly used as the pixel scale reference over*

*homogeneous surfaces or in the case that the satellite acquisition and in situ measurement footprints are similar, and the upscaling model is not necessary as it has its own source of uncertainty. But the upscaling model is useful for heterogeneous areas when in situ measurement footprints are less than satellite pixel*
170 *size........".*

Lines 55-56: The community-agreed surface albedo validation protocol (CEOS Working Group on Calibration and Validation – Land Product Validation subgroup) disagreed with this affirmation. Ground measurement can be directly used to validate satellite pixels. The current community-agreed approach is
175 based on the evaluation of the spatial representativeness of ground measurement (Román et al., 2010, 2009). Reference: CEOS LPV albedo protocol: https://lpvs.gsfc.nasa.gov/PDF/CEOS_ALBEDO_Protocol_20190307_v1.pdf.

Re: It is true that the ground measurement can be directly used to validate satellite pixels after proving that *in situ* measurements are spatially representative. However, the representative site are only limited to a
180 few locations on the globe and cover discrete time periods, which cannot support a comprehensive validation and bias correction over a wide range of conditions (Loew et al., 2016). Upscaling procedure is necessary for heterogeneous areas when *in situ* measurement footprints are less than satellite pixel size. Hence, our dataset can be considered as an important addition to the reference data on the coarse pixel scale. In order to clarify this point, we have added the sentence as "*Currently, a community-based validation tool,*
185 *such as SALVAL (Sánchez-Zapero et al., 2023), could provide a framework for undertaking performance assessments through well-defined and uniform procedures, metrics and reference observations for all involved datasets, resulting in increased comparability, in addition to the ability to import new product datasets. Our dataset, obtained through standardized operational procedures, permits expanding established datasets to spatially underrepresented sites.*" in **Conclusion**.

190

Lines 56-57: 'Limited by the means and methods of ground measurement, the absolute truth on the coarse pixel scale cannot be obtained.' Justify this sentence.

Re: This sentence has been removed from the paper. Instead, this point has been clarified in
195 **Conclusion** as "*It is important to note that the absolute truth on the coarse pixel scale is unattainable due to the limitations in instruments and measurement methods as well as the uncertainty of the upscaling model (Wu et al., 2019; Wen et al., 2022). Instead, the relative truth can be used to approximate the absolute truth.*".

The reason that the absolute truth on the coarse pixel scale being unattainable can be explained from
200 the following aspects. First, *in situ* measurements inevitably suffer from errors (random errors and systematic errors). The systematic errors can be corrected through calibration. While the random error can be reduced with repeated measurements, the repeatability in the exactly same conditions is hard to implement in the natural environment. Second, the scale of *in situ* measurements is generally less than

satellite pixel size and lacks representativeness due to spatial heterogeneity. Third, the upscaling procedure suffers from its own source of uncertainty.

Lines 65-67: 'However, *in situ* measurements cannot be directly used as the coarse pixel scale truth given that the footprint of *in situ* sites is far less than the scale of a coarse pixel.' Please justify this or rephrase this sentence. *In situ* measurement footprint depends on the tower height. Depending of tower height and satellite spatial resolution they can be compared.

Re: This sentence has been rephrased as "*However, in situ measurements cannot be directly used as the coarse pixel scale truth if the footprint of in situ sites (depending on tower height) is far less than the scale of a coarse pixel.*".

Lines 115-118: 'These radiometers have been rigorously calibrated and continuously supervised to reduce systematic measurement errors (Jia et al., 2013; Wang et al., 2009; Zhou et al., 2016).' Are you confident that all radiometers from 368 sites have been rigorously calibrated and continuously supervised? This is not the case based on the references you are providing.

Re: In fact, most of these radiometers have been rigorously calibrated and continuously supervised. To remove the effect of *in situ* measurement uncertainty caused by the lack of strict calibration or supervision, we have made a quality control of *in situ* measurements. The outliers have been removed. Furthermore, the possible effects of unstable lighting on flux measurements were also minimized by using the ratio of the mean upward radiation to the mean downward radiation around local solar noon (11:00–13:00) as suggested by Lin et al. (2022). In order to clarify this, we have added the sentence as "*To reduce the possible effects of unstable lighting on flux measurements and align with satellite albedo products that generally report local solar noon albedo, in situ site measured albedo was calculated using the ratio of the mean upward radiation to the mean downward radiation around local solar noon (11:00–13:00) as suggested by Lin et al. (2022).*" in **Section 2.1**.

Lines 118-120: Justify the use of measurement at the local solar noon.

Re: The reasons for using measurement at the local solar noon are as follows:

First, satellite albedo products such as MCD43A3 V061 typically provide local noon solar albedo;

Second, surface albedo (especially black-sky albedo) is sensitive to the sun zenith angle, and the temporal variation of surface albedo around local solar noon is minimal, which is helpful for the temporal match between *in situ* and satellite measurements.

To clarify this point, the corresponding part has been revised as "*To reduce the possible effects of unstable lighting on flux measurements and align with satellite albedo products that generally report local solar noon albedo, in situ site measured albedo was calculated using the ratio of the mean upward radiation to the mean downward radiation around local solar noon (11:00－13:00) as suggested by Lin et al. (2022).*" in **Section 2.1**.

The formula proposed to combine WSA and BSA used the diffuse light ratio, which is an approximation. The actual diffuse solar radiation should be used, as is the real model considering the actual environment (as you said), as considers the real atmospheric state.

Justify the use of this approximation, including the uncertainties introduced in this step. The limitations over snow targets should be also discussed.

I cannot find the formula used to calculate sky diffuse light ratio in the provided reference (Stokes and Schwartz (1994)). Please, use the right reference.

Re: The use of this approximation can be explained from the following aspects:

First, the *in situ* sites used in this paper cover a wide range of environmental conditions (geographic locations, atmospheric model, aerosol model, spatial homogeneity and heterogeneity, temporal variation characteristics). Hence, the input parameters for the physical models such as 6S and MODTRAN are difficult to be precisely set.

Second, the formula we employed is simple, in which the sky diffuse light ratio is merely a function of the solar zenith angle at local solar noon. Hence, it can be applied to all of these *in situ* sites.

Third, although the formula is an empirical function, it has been widely accepted and used in previous studies (An et al., 2022; Mao et al., 2022; Wang et al., 2014; Lewis and Barnsley, 1994). These right references have been used in the revised manuscript.

Regarding the limitations over snow targets, it is true that the underlying assumption of an isotropic distribution of the diffuse skylight cannot be fully satisfied, but it avoids the expense of an exact calculation while capturing the major part of the phenomenon (Pinker and Laszlo, 1992). Moreover, Lucht et al. (2000) also pointed out that the fraction of diffuse to total irradiation can be parameterized in a relatively simple way at least for moderate solar zenith angles. In order to clarify this point, we have added the sentence as "*In this study, we approximated the proportion of diffuse irradiation as a function of the cosine of the solar zenith angle at noon using an empirical statistical equation (i.e., Eq. (3)). Although this equation is approximate, it avoids the excessive amount of calculation while capturing the major phenomenon (Pinker and Laszlo, 1992). In fact, this empirical function has been widely used by previous studies (An et al., 2022; Mao et al., 2022; Wang et al., 2014b; Lewis and Barnsley, 1994).*" in **Section 2.3**.

**References:**

An, Y., Meng, X., Zhao, L., Li, Z., Wang, S., Shang, L., Chen, H., and Lyu, S.: Evaluation of surface albedo over the Tibetan Plateau simulated by CMIP5 models using in-situ measurements and MODIS. International Journal of Climatology, 42(2), 928–951, https://doi.org/10.1002/joc.7281, 2022.

Mao, T., Shangguan, W., Li, Q., Li, L., Zhang, Y., Huang, F., Li, J., Liu, W., and Zhang, R.: A Spatial Downscaling Method for Remote Sensing Soil Moisture Based on Random Forest Considering Soil Moisture Memory and Mass Conservation, Remote Sensing, 14, 3858, https://doi.org/10.3390/rs14163858, 2022.

275   Wang, L., Zheng, X., Sun, L., Liu, Q., and Liu, S.: Validation of GLASS albedo product through Landsat TM data and ground measurements, Journal of Remote Sensing, 18(3), 547-558, https://doi.org/10.11834/jrs.20143130, 2014.

Lewis, P., and Barnsley, M. J.: Influence of the sky radiance distribution on various formulations of the Earth surface albedo, International Symposium on Physical Measurements and Signatures in Remote Sensing, 17-22, 707-715, available at: http://www2.geog.ucl.ac.uk/~plewis/LewisBarnsley1994.pdf (last access: 23 September 2023), 1994.

280   Pinker, R. T., and Laszlo, I.: Modeling Surface Solar Irradiance for Satellite Applications on a Global Scale, Journal of Applied Meteorology and Climatology, 31(2), 194-211, https://doi.org/10.1175/1520-0450(1992)031<0194:MSSIFS>2.0.CO;2, 1992

Lucht, W, Schaaf, C. B., and Strahler, A. H.: An algorithm for the retrieval of albedo from space using semiempirical BRDF models, IEEE Transactions on Geoscience and Remote Sensing, 38(2), 977-998,
285   https://doi.org/10.1109/36.841980, 2000.

Not clear what definition of satellite product is used according to illumination geometry (black-sky, white-sky)? Please provide more details about that.

Re: The blue-sky albedo which encompasses both direct and diffuse components and denotes the land surface albedo under actual atmospheric conditions, was used in this study.

290   The MCD43A3 V061 product was used as an example of coarse-resolution satellite albedo products. This product provides local solar noon black sky albedo (BSA) and white sky albedo (WSA). The blue-sky albedo under the actual environment can be calculated as a linear combination of BSA and WSA through the proportion of diffuse irradiation. To clarify this point, we have revised the sentence as "*The blue-sky albedo encompasses both direct and diffuse components, characterizing the albedo of the surface under*
295   *actual atmospheric conditions. It can be expressed as a linear combination of BSA and WSA with an assumption of isotropic distribution of diffuse radiation. In this study, the following equation is used to calculate the MODIS blue-sky albedo.....*" in **Section 2.3.**

The Landsat ETM+ albedo was used as an example of high-resolution albedo products. The method we employed directly calculates the blue-sky albedo. For clarification, we have revised the sentence as "*In this*
300   *study, we employed the following equation to calculate shortwave blue-sky albedo estimates.*" in **Section 2.1**.

lines 177-189: This part does not correspond to ancillary data. Here you are describing the spatial heterogeneity metric (std) that should be moved to the 'methodology' section.

305   Re: As suggested by the reviewer, the description of spatial heterogeneity metric (std) has been moved to the methodology section (i.e., **Section 3.2.3**).

I miss a diagram clearly showing the process of the upscaling model.

Re: The process of the upscaling method is shown as follows.

310

[Figure]

**Figure. Framework of the upscaling method.**

However, since the paper was focused on the comprehensive evaluation of the upscaling model and the development of the pixel scale ground "truth" dataset, the flowchart of the upscaling method itself was not shown in the revised manuscript.

The performance of the upscaling model shows that the uncertainty (RMSE) of the upscaled maps is typically between 0.03 and 0.05, which is the typical uncertainty of the surface albedo coarse resolution satellite products (e.g., MCD43A3, GLASS, GlobAlbedo, C3S SPOT/VGT, C3S PROBA-V, C3S Sentinel-3). In conclusion, the uncertainty of the upscaled maps is similar to any other product and it is questionable its utility as a reference 'ground-truth'.

Re: It is true that the upscaling model itself has errors because it suffers from its own source of uncertainty. Therefore, over homogeneous surfaces where *in situ* site measurements are spatially representative, using this upscaling model will bring no benefits or even counteract due to the errors of the upscaling model. Nevertheless, over the heterogeneous surface where *in situ* sites are lack of spatial representativeness, the benefits outweigh disadvantages. The accuracy assessment results of the coarse pixel scale ground "truth" dataset demonstrate that the accuracy of reference data can be enhanced by 17.09 % over the regions with strong spatial heterogeneity. However, the degree of improvement with this dataset displays a decreasing trend as the reduction of spatial heterogeneity. In order to clarify this point, we have added the paragraph "*……For instance, the in situ measurements can be directly used as the pixel scale reference over homogeneous surfaces or in the case that the satellite acquisition and in situ measurement footprints are similar, and the upscaling model is not necessary as it has its own source of uncertainty. But the upscaling model is useful for heterogeneous areas when in situ measurement footprints are less than*

*satellite pixel size, because it increases the representativeness of the sampling for direct validation. The*
335 *accuracy assessment results of pixel scale ground "truth" dataset demonstrate that the accuracy of*
*reference data can be enhanced by 17.09 % over the regions with strong spatial heterogeneity……"* in
**Conclusion**.

As regards to the accuracy of the current coarse resolution surface albedo satellite products, their
accuracy (between 0.03 and 0.05) is usually assessed over relatively homogeneous land surfaces. And the
340 validation works over heterogeneous are still rare currently. The spatial scale mismatch over heterogeneous
surfaces remains to be challenging to fully understand the overall accuracy of satellite products in different
areas. Hence, our dataset can be considered as an important addition to the reference data on the coarse
pixel scale over heterogeneous land surfaces.

345 It is not clear which albedo quantities are you comparing: albedo single site, albedo upscaling, reference?
You should focus your discussion also based on the different albedo definitions of these quantities (blue-sky,
black-sky, etc). It is not clear the spatial coverage of the study. You should clearly indicate the spatial
resolution related to all datasets used in this section: albedo single site, albedo upscaling, reference.

Re: We are sorry for not making it clear to readers. In fact, it was blue-sky albedo that was used in this
350 study.

The MCD43A3 V061 product was used as an example of coarse resolution satellite albedo products.
This product provides local solar noon black sky albedo (BSA) and white sky albedo (WSA). The blue-sky
albedo under the actual environment can be calculated as a linear combination of BSA and WSA through
the proportion of diffuse irradiation. To clarify this point, we have revised the sentence as "The blue-sky
355 albedo encompasses both direct and diffuse components, characterizing the albedo of the surface under
actual atmospheric conditions. It can be expressed as a linear combination of BSA and WSA with an
assumption of isotropic distribution of diffuse radiation. In this study, the following equation is used to
calculate the MODIS blue-sky albedo….." in **Section 2.3**.

The Landsat ETM+ albedo was used as an example of high-resolution albedo products. The method we
360 employed directly calculates the blue-sky albedo. For clarification, we have revised the sentence as "In this
study, we employed the following equation to calculate shortwave blue-sky albedo estimates." in **Section
2.1**.

*In situ* blue-sky albedo was calculated using the ratio of the mean upward radiation to the mean
downward radiation around local solar noon. To make this clear to readers, we have added the "*blue-sky*" in
365 **Section 2.1**.

Regarding the spatial coverage of the study, the *in situ* sites are globally distributed as shown in Figure
1. The spatial resolution related to all datasets has been summarized as tables in the above.

The validation of MCD43A3 V0061 using pixel scale ground 'truth' is only presented for some sites. The

selection of these sites (and not others) should be justified. What is the reason of large differences (outliers) over CA-NS2, CA-LP1, IT-Tor ? Additionally, I miss the overall figure using the whole dataset.

Re: In fact, the validation of MCD43A3 V0061 was merely presented as an example for the usage of pixel scale ground "truth". Only parts of the sites were shown for conciseness. These sites are selected randomly for each land cover type with consideration of different degrees of spatial heterogeneity. The overall figure was not shown since the focus of this paper is not comprehensively assess the accuracy of satellite albedo products.

There already exist other initiatives, like GBOV (https://gbov.acri.fr/), providing similar datasets to that presented in this manuscript, and should be mentioned.

On the other case, during the manuscript there are comments related to lack of standardized methods and operational validation systems for albedo validation. In fact, the CEOS/WGCV LPV subgroup (https://lpvs.gsfc.nasa.gov/) is coordinating these activities. An operational validation system was recently endorsed by CEOS/WGCV LPV, which is called SALVAL (Sánchez-Zapero et al., 2023) and it allows albedo products to reach operational and globally representative validation results (CEOS LPV stage 4). Access to SALVAL is available on https://calvalportal.ceos.org/salval

Re: Great thanks for the comment. As suggested by the reviewer, we have added a comment about the exsiting datasets and validation activities in **Introduction** as "*It is important to note that the Copernicus Global Terrestrial Monitoring Service partners have instituted a centralized validation database known as the Copernicus Global Terrestrial Product Validation Ground-based Observation Dataset (GBOV, http://gbov.copernicus.acri.fr), providing direct access to the set of reference measurements. However, the Copernicus GBOV ground-based observation dataset merely comprises 20 stations that provide albedo reference data, and the scope of these reference data is inadequate to systematically evaluate remote sensing products globally. Thus, our collection of ground-based "truth", which covers the widest spatial range and the longest time series on the coarse pixel scales, is essential to supplement the scientific efforts on existing albedo datasets and deliver a more precise and consistent albedo reference dataset on the coarse pixel scale for heterogeneous regions.*" and **Conclusion** as "*Currently, a community-based validation tool, such as SALVAL (Sánchez-Zapero et al., 2023), could provide a framework for undertaking performance assessments through well-defined and uniform procedures, metrics and reference observations for all involved datasets, resulting in increased comparability, in addition to the ability to import new product datasets. Our dataset, obtained through standardized operational procedures, permits expanding established datasets to spatially underrepresented sites.*".

Line 18: 'in situ' is not hyphenated. Please review the whole manuscript to homogenize 'in situ' term.

Re: This has been corrected in the revised manuscript.

Line 64: Remove '.' before references

Re: I' ve revised the mistake in the article:

Line 145: 'ith' ?

410 Re: 'ith' typically represents a specific index or instance, For example, '$\alpha_5$' might denote the fifth satellite spectral band.

---

## Referee Report (RR1)

2nd Review of "A coarse pixel scale ground "truth" dataset based on the global in situ site measurements to support validation and bias correction of satellite surface albedo products" by Fei Pan et al.

The new manuscript has been revised greatly. Authors should answer the following comments before the publication.

**Major comments:**
1. Lines 287-288: It's difficult to see "BSRN network generally exhibits higher accuracy and satisfies the precision benchmarks" in Figure 5. Please show more analysis (Figures or tables as you like) to support your point.
2. Lines 297-298: Authors highlighted that "It is worth noting that when the spatial heterogeneity exceeds 0.1, the model's stability fluctuates considerably, indicated by the larger height of the boxplots of RMSE and R2." However, the height of the boxplots of RMSE with spatial heterogeneity < 0.1 in Figure 6a is much larger than that of the other two obviously, while the outliers with spatial heterogeneity < 0.1 in both panels of Figure 6 are much more than those of the other two significantly. Therefore, how can the authors highlight the above result?
3. Figure 11: Please explain why the results (percentages) in Figure 11 is quite different in the revised manuscript than those in the previous manuscript (version 1)? In previous version, the RRMSEs are quite lower (around 30%) than those (around 100%) in the revised version.

**Minor comments:**
1. Section 2.2: the description of ETM+ data should be described in detail. I cannot find the resolution of the data here which should not be mentioned in section 3.2. Please modify.
2. Please check the caption of Figure 7. Duplicated [200-500].
3. The contents of functions are overlapping in the PDF version. Please double-check the typing of all functions.

---

## Author Response (AR2)

**Response to comments**

**Paper #:** essd-2023-220

**Title:** A coarse pixel scale ground "truth" dataset based on the global in situ site measurements to support validation and bias correction of satellite surface albedo products

**Journal:** Earth System Science Data

We really appreciate the rigorous attitude of the reviewer for providing so many valuable suggestions. We revised the paper carefully and tried to give satisfactory answers to the reviewers' questions. The corresponding modifications are highlighted in red font in the revised paper.

First, we have added the diagram showing the upscaling and evaluation process.

Second, we have emphasized the intention and objective of generating such a pixel scale ground albedo "truth" dataset, and further explained the uncertainty of the upscaling model.

Third, we have plotted the distribution of the accuracy indicators for different networks and rephrased the sentence about the performance of BSRN network.

Fourth, we have explained that only the median values of the boxplots were focused due to the even sample sizes for each level of spatial heterogeneity.

Fifth, the reason why the results (percentages) in Figure 11 is quite different in the revised manuscript than those in the previous manuscript was given in this manuscript: the addition of new in situ sites and the the different data sources of the reference data.

**Reviewer #1**

I would still suggest adding the diagram showing the upscaling and evaluation process but avoiding plagiarism.

Re: We sincerely appreciate your rigorous science attitude. As you suggested, we have added and improved the diagram by adding the procedure of evaluation process of the upscaling model and the generation of the pixel scale reference albedo dataset based on the upscaling model and *in situ* site measurements in the revised manuscript. The revised diagram is as follows:

[Figure]

Figure R1: The flowchart of generating coarse pixel scale ground "truth" based on upscaling model.

The method part should emphasize the improvements compared with the published methods on developing a global pixel scale ground "truth" dataset.

Re: In fact, this paper is the continuation and deepening of our previous work. The upscaling method was proposed in our previous research, but the effectiveness of this upscaling method on the global scale was still unknown. Furthermore, this method has not been utilized for individual *in situ* site measurements from sparsely globally distributed observation networks (e.g., SURFRAD, BSRN, and Fluxnet). Under this background, this study aims to first comprehensively evaluate the effectiveness of upscaling methods on the global scale, and then apply this upscaling method to the 416 in situ sites over the globe. Finally, a pixel scale ground "truth" dataset was provided for validation, bias correction, and other applications that need the linkage between *in situ* measurements and satellite pixels.

Besides, I recommend the author delve deeper into the previous comments 'the uncertainty of the upscaled maps as similar to any other product and it is questionable its utility as a reference ground-truth'.

Re: Thank you for your insightful comments. It's important to note that the accuracy (between 0.03 and 0.05) of current coarse-resolution surface albedo satellite products was generally assessed over relatively homogeneous land surfaces. But their accuracy over heterogeneous are still unknown, because the effect of scale mismatch between *in situ* measurements and satellite pixel cannot be ignored but not resolved. The scale mismatch is still the key challenge over heterogeneous surface. And this is original intention of generating such a pixel scale ground truth dataset. Moreover, we would like to point out that the accuracy of the pixel scale ground truth cannot be determined through the comparison with other products since the products themselves contain errors. In fact, the advantage of the pixel scale ground truth was proved

through the comparison with in situ single site measurements in terms of their agreements with a coarse

50     pixel scale albedo value. Therefore, although the pixel scale ground truth dataset is not the absolute truth due to its own uncertainty, it shows an advantage over single *in situ* sites when matched with satellite pixel.

**Reviewer #2**

2nd Review of "A coarse pixel scale ground "truth" dataset based on the global in situ site measurements to

55     support validation and bias correction of satellite.

2nd Review of "A coarse pixel scale ground "truth" dataset based on the global in situ site measurements to support validation and bias correction of satellite surface albedo products" by Fei Pan et al.

The new manuscript has been revised greatly. Authors should answer the following comments before the publication.

60     **Major comments:**

1. Lines 287-288: It's difficult to see "BSRN network generally exhibits higher accuracy and satisfies the precision benchmarks" in Figure 5. Please show more analysis (Figures or tables as you like) to support your point.

      Re: We are so sorry for not explaining this clearly. To illustrate this point, we have plotted the

65     distribution of the accuracy indicators for different networks (Figure R2). It can be seen that the advantage of BSRN is not significant compared to other networks given that their RMSE and $R^2$ are comparable. To clarify this point, the sentence has been rephrased as "*Both GCOS and CEOS LPV albedo best practice protocols (Wang et al., 2019) indicate the better performance of BSRN than other networks. However, this phenomenon does not occur with this upscaling model given the comparable RMSE and $R^2$ among different*

70     *networks*" in the revised manuscript.

[Figure]

**Figure R2. Distribution of RMSE (a), and R² (b) of the five Networks used in the study.**

2. Lines 297-298: Authors highlighted that "It is worth noting that when the spatial heterogeneity exceeds 0.1, the model's stability fluctuates considerably, indicated by the larger height of the boxplots of RMSE and R2." However, the height of the boxplots of RMSE with spatial heterogeneity < 0.1 in Figure 6a is much larger than that of the other two obviously, while the outliers with spatial heterogeneity < 0.1 in both panels of Figure 6 are much more than those of the other two significantly. Therefore, how can the authors highlight the above result?

Re: We appreciate the reviewer's careful observation and comment. In fact, the data points beyond the upper and lower edges of the boxplots were identified as the outliers, and this is just the unique advantage of boxplots. These outliers should be excluded from the analysis. Generally, the median value as well as the interquartile range should be the measures of performance. However, since the sample sizes are not equal under various spatial heterogeneity conditions, only the median values were focused in this study because it is less influenced by sample size.

The sentence "*It is worth noting that when the spatial heterogeneity exceeds 0.1, the model's stability fluctuates considerably, indicated by the larger height of the boxplots of RMSE and $R^2$*." was not enough rigorous and thus was revised as "*It is worth noting that when the spatial heterogeneity exceeds 0.1, the $R^2$ of the model fluctuates considerably, indicated by the larger height of the boxplots*" in the revised manuscript.

3. Figure 11: Please explain why the results (percentages) in Figure 11 is quite different in the revised manuscript than those in the previous manuscript (version 1)? In previous version, the RRMSEs are quite

lower (around 30%) than those (around 100%) in the revised version.

95      Re: The difference in the RRMSEs between this version and last version was caused by two reasons. First, the addition of new in situ sites. In the revised version, we have incorporated a significant amount of new site data into our analysis. The introduction of these new sites led to differences between the overall accuracy and performance of the current version and the previous version. Second, the different data sources of the reference data. In the previous version, the MODIS albedo product was used as the reference.

100     However, in the current version, the aggregated Landsat ETM+ albedo on the 500 m pixel scale was used as the reference.

        For comparison purpose, we have plotted the RRMSE based on the MODIS albedo product (Figure R3) and the aggregated HJ albedo (Figure R4) over the 416 in situ sites , respectively. It can be seen that although the RRMSE present large difference when different data was used as the reference, the RRMSE of

105     pixel scale ground "truth" were always smaller than the single in situ site measurements, demonstrating the advantage of pixel scale ground truth over single in situ site measurements. It is important to note that the absolute accuracy of the pixel scale ground truth cannot be determined through comparison with other products (e.g., MODIS albedo or ETM+ albedo) since the products themselves contain errors. Instead, the advantage of the pixel scale ground truth was proved through the comparison with in situ single site

110     measurements in terms of their relative accuracy relative to a coarse pixel scale albedo value (e.g., MODIS albedo or ETM+ albedo). In other words, the value of RRMSE was not the focus, but the difference of RRMSEs between the pixel scale ground truth and single in situ site measurements was the key. In order to clarify this point, we have added the sentence "*Although the errors of the pixel scale ground "truth" are not negligibly small, it is important to note that this kind of error cannot reveal the absolute accuracy of pixel*

115     *scale ground "truth" given that the reference data itself contain errors. In fact, the focus of this evaluation is not the value of RRMSEs but the difference of RRMSEs between the pixel scale ground "truth" and single in situ site measurements. It can be seen that the accuracy of the pixel scale ground "truth"……*" in **Section 4.2**. Since the previous studies (Peng et al., 2015; Wen et al., 2022, Wu et al., 2016) generally used the aggregated high-resolution albedo as the reference on the coarse pixel scale, we employed the aggregated

120     Landsat ETM+ albedo in this version to be consistent with previous studies.

[Figure]

**Figure R3**: The boxplots of RRMSE of pixel scale ground "truth" and single site measurements. The reference data was MODIS albedo product.

[Figure]

**Figure R4**: The boxplots of RRMSE of pixel scale ground "truth" and single site measurements. The reference data was aggregated ETM+ albedo.

**Minor comments:**

1. Section 2.2: the description of ETM+ data should be described in detail. I cannot find the resolution of the data here which should not be mentioned in section 3.2. Please modify.

Re: The resolution of ETM+ imagery bands has been mentioned in Section 3.2.1 as "……*a critical component of the upscaling approach involves the acquisition of upscaling coefficients derived from 30-meter ETM+ albedo covering the period from 2012 to 2018*".

2. Please check the caption of Figure 7. Duplicated [200-500].

135       Re: We have corrected these errors in the revised manuscript.

3. The contents of functions are overlapping in the PDF version. Please double-check the typing of all functions.

      Re: We are sorry for this mistake. We have carefully reviewed the revised manuscript and have made
140   sure that the formatting and typing of all functions are correct.